# Diffusion Bridge Variational Inference for Deep Gaussian Processes

**Jian Xu**
School of Mathematics
South China University of Technology &
RIKEN AIP
jian.xu@riken.jp

**Delu Zeng** *
School of Electronics and Information Engineering
South China University of Technology &
Department of Electrical and Computer Engineering
University of Waterloo
dlzeng@scut.edu.cn

**Qibin Zhao**
RIKEN AIP
qibin.zhao@riken.jp

**John Paisley**
Columbia University
jwp2128@columbia.edu

## Abstract

Deep Gaussian processes (DGPs) enable expressive hierarchical Bayesian modeling but pose substantial challenges for posterior inference, especially over inducing variables. Denoising diffusion variational inference (DDVI) addresses this by modeling the posterior as a time-reversed diffusion from a simple Gaussian prior. However, DDVI's fixed unconditional starting distribution remains far from the complex true posterior, resulting in inefficient inference trajectories and slow convergence. In this work, we propose Diffusion Bridge Variational Inference (DBVI), a principled extension of DDVI that initiates the reverse diffusion from a learnable, data-dependent initial distribution. This initialization is parameterized via an amortized neural network and progressively adapted using gradients from the ELBO objective, reducing the posterior gap and improving sample efficiency. To enable scalable amortization, we design the network to operate on the inducing inputs $\mathbf{Z}^{(l)}$, which serve as structured, low-dimensional summaries of the dataset and naturally align with the inducing variables' shape. DBVI retains the mathematical elegance of DDVI—including Girsanov-based ELBOs and reverse-time SDEs—while reinterpreting the prior via a Doob-bridged diffusion process. We derive a tractable training objective under this formulation and implement DBVI for scalable inference in large-scale DGPs. Across regression, classification, and image reconstruction tasks, DBVI consistently outperforms DDVI and other variational baselines in predictive accuracy, convergence speed, and posterior quality.

## 1 Introduction

Deep Gaussian processes (DGPs) (Damianou & Lawrence, 2013) extend the representational capacity of Gaussian processes (GPs) (Seeger, 2004) by composing multiple layers of latent functions, enabling flexible hierarchical Bayesian modeling. However, posterior inference in DGPs is notoriously challenging due to the non-conjugate likelihoods, strong inter-layer dependencies, and the large number of inducing variables required for scalability. Stochastic variational inference (SVI) with inducing points (Hensman et al., 2013; Salimbeni & Deisenroth, 2017; Xu & Zeng, 2024) has become the standard approach, but designing accurate and efficient variational posteriors remains a central bottleneck.

Recently, denoising diffusion variational inference (DDVI) (Xu et al., 2024c) has been proposed to approximate the inducing-point posterior via the time-reversal of a diffusion stochastic differential equation (SDE) (Song et al., 2020) starting from a simple Gaussian prior. By parameterizing the reverse drift with a neural network, DDVI can flexibly capture complex posteriors while retaining

---
*Corresponding author.

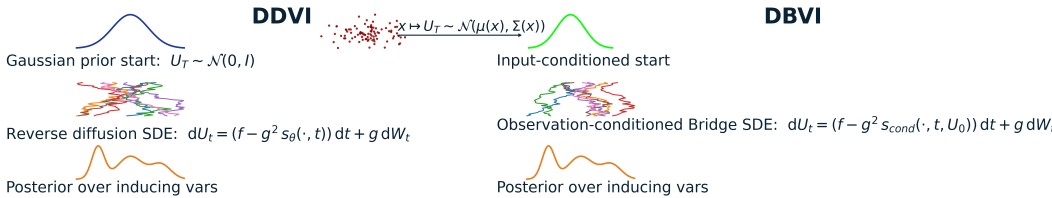

Figure 1: **Comparison between DDVI and DBVI.** (Left) DDVI starts from an unconditional Gaussian prior and runs a reverse diffusion SDE towards the posterior. (Right) DBVI starts from an input-conditioned initial distribution and uses an observation-conditioned diffusion bridge SDE, leading to shorter and more efficient inference trajectories.

scalable SVI (Hoffman et al., 2013; Xu et al., 2024b; Chen et al., 2026b) training. However, a key limitation of DDVI lies in its reliance on a fixed, unconditional Gaussian distribution as the start of the reverse diffusion. Since the true posterior over inducing variables is typically far from this initial distribution, the reverse-time SDE must traverse a long and complex path to reach the target, resulting in inefficient inference and slow convergence.

To address this, in this work, we propose **Diffusion Bridge Variational Inference (DBVI)**, which replaces the unconditional reverse diffusion in DDVI with a *learnable, input-conditioned distribution* that adapts over training. By parameterizing the start point of the diffusion using a neural network, gradients from the ELBO naturally push the initial distribution closer to the posterior. This effectively narrows the inference gap and alleviates the burden on the reverse SDE, yielding more stable and sample-efficient inference. Moreover, conditioning the initial distribution on observations introduces a natural connection to amortized variational inference: the generative structure of DBVI allows each posterior sample to be generated by a single forward pass of a drift network, without requiring per-dataset optimization. Unlike naive amortization that conditions directly on raw inputs—leading to high-dimensional mismatches and overfitting—our design uses the inducing inputs $\mathbf{Z}^{(l)}$ at each layer as input proxies to the amortizer. This enables batch-wise inference while preserving global dataset structure and matching the dimensionality of inducing variables.

Importantly, the theoretical elegance of DDVI—particularly its use of time-reversed SDEs, Girsanov's theorem (Li et al., 2020), and ELBO construction—remains fully preserved in DBVI. We build upon the same diffusion framework, but reinterpret the prior as a *Doob-bridged process* whose start is parameterized by an amortized network. This allows DBVI to inherit the key benefits of DDVI while significantly improving its flexibility and inference efficiency. We derive an evidence lower bound (ELBO) objective for training DBVI within the SVI framework and provide a scalable implementation for large-scale DGPs. Empirical results on regression, classification, and image reconstruction benchmarks demonstrate that DBVI achieves more accurate posterior approximations, faster convergence, and improved predictive performance compared to DDVI and other state-of-the-art DGP inference methods. Our contributions are as follows:

- We propose Diffusion Bridge Variational Inference (DBVI), a novel extension of DDVI that replaces the unconditional start of the reverse diffusion with a learnable, input-conditioned initial distribution, effectively reducing the inference gap and improving posterior approximation.

- We introduce a bridge-based reinterpretation of the DDVI framework by integrating Doob's h-transform into the variational formulation, while preserving the core machinery of reverse-time SDEs, Girsanov-based ELBO, and SVI scalability.

- We develop a structured amortization strategy that leverages inducing locations $\mathbf{Z}^{(l)}$ as the input to the drift network, enabling batch-wise amortized inference without requiring access to the full dataset or high-dimensional conditioning on raw inputs.

- We provide a scalable implementation of DBVI for deep Gaussian processes and validate its performance on regression, classification, and image reconstruction tasks, showing consistent improvements over DDVI and other state-of-the-art DGP inference methods in terms of accuracy, convergence speed, and sample efficiency.

## 2 RELATED WORK

**Deep Gaussian Process Inference.** Scalable DGP inference initially extended sparse variational and inducing-point GP methods to multilayer settings (Hensman et al., 2013; Damianou & Lawrence, 2013; Salimbeni & Deisenroth, 2017; Xu et al., 2024a;c; 2025a;b), with DSVI improving practicality (Salimbeni & Deisenroth, 2017) but still seeming to struggle to capture complex inducing-variable posteriors, particularly in deeper models. IPVI (Yu et al., 2019) addresses expressiveness by using a neural network to represent the inducing-point posterior and training it via an adversarial, GAN-like objective (Goodfellow et al., 2014); however, this formulation appears hard to optimize in practice and can be unstable, which may lead to biased posterior estimates.

**Diffusion-based Variational Inference.** Score-based generative modeling and diffusion probabilistic models (Song et al., 2020; Ho et al., 2020; Li et al., 2023b; Chen et al., 2024; Li et al., 2025b;a; Chen et al., 2026a) have inspired new VI methods (Xu et al., 2024c) that represent posteriors as solutions to reverse-time SDEs. DDVI (Xu et al., 2024c) applies this to DGP inference, parameterizing the reverse drift via a score network. However, its unconditional start distribution can be far from the posterior, requiring long diffusion paths and increasing variance. Our DBVI can be viewed as a strict extension of DDVI: same theoretical framework, but with bridge correction and amortized initialization, yielding both theoretical guarantees and empirical improvements.

**Diffusion Bridge Models.** Diffusion bridges (Zhou et al., 2023; Li et al., 2023a; Lin et al., 2025; Chen et al., 2025) constrain dynamics between fixed endpoints or distributions, enabling more direct and sample-efficient transitions. Observation-conditioned bridges have been explored in Schrödinger bridge formulations (Shi et al., 2023) and consistency diffusion models (He et al., 2024). Our DBVI adapts this idea to variational inference in DGPs, integrating an *amortized* parameterization (Kim et al., 2018; Agrawal & Domke, 2021; Margossian & Blei, 2023; Ganguly et al., 2023) to map inputs to initial states, reducing the KL gap and improving efficiency.

## 3 METHOD

### 3.1 DEEP GAUSSIAN PROCESSES

Deep Gaussian Processes (DGPs) (Damianou & Lawrence, 2013) generalize standard Gaussian Processes (GPs) by hierarchically composing multiple GP layers, enabling deep non-linear probabilistic mappings. Let $\mathbf{x} \in \mathbb{R}^d$ be an input and $\mathbf{y} \in \mathbb{R}^p$ the corresponding output. A DGP with $L$ layers defines latent variables $\{\mathbf{f}^{(l)}\}_{l=1}^L$ recursively through:

$$p\big(\mathbf{f}^{(l)} \mid \mathbf{f}^{(l-1)}\big) = \mathcal{GP}\Big(0, k^{(l)}\big(\mathbf{f}^{(l-1)}, \mathbf{f}^{(l-1)}\big)\Big), \tag{1}$$

where $k^{(l)}$ is the kernel function at layer $l$ with hyperparameters $\boldsymbol{\gamma}^{(l)}$. For scalability, each layer introduces $M_l$ inducing variables $\mathbf{u}^{(l)}$ located at inducing inputs $\mathbf{Z}^{(l)}$, with GP prior:

$$p_{\text{prior}}\big(\mathbf{u}^{(l)}\big) = \mathcal{N}\big(\mathbf{0}, \mathbf{K}_{\mathbf{ZZ}}^{(l)}\big). \tag{2}$$

Assuming conditional independence across layers given inducing variables, the full joint distribution over outputs $\mathbf{y}$, latent variables $\{\mathbf{f}^{(l)}\}$, and inducing variables $\{\mathbf{u}^{(l)}\}$ factorizes as:

$$p(\mathbf{y}, \mathbf{F}, \mathbf{U}) = \left[\prod_{l=1}^L p\big(\mathbf{f}^{(l)} \mid \mathbf{f}^{(l-1)}, \mathbf{u}^{(l)}\big)\, p\big(\mathbf{u}^{(l)}\big)\right] p\big(\mathbf{y} \mid \mathbf{f}^{(L)}\big), \tag{3}$$

where the inter-layer conditional distribution $p\big(\mathbf{f}^{(l)} \mid \mathbf{f}^{(l-1)}, \mathbf{u}^{(l)}\big)$ is given by the sparse GP conditional distribution

$$\begin{aligned} &p\big(\mathbf{f}^{(l)} \mid \mathbf{f}^{(l-1)}, \mathbf{u}^{(l)}\big) \\ &= \mathcal{N}\left(\mathbf{K}_{\mathbf{f}^{(l-1)}\mathbf{Z}^{(l)}}\mathbf{K}_{\mathbf{Z}^{(l)}\mathbf{Z}^{(l)}}^{-1}\mathbf{u}^{(l)},\ \mathbf{K}_{\mathbf{f}^{(l-1)}\mathbf{f}^{(l-1)}} - \mathbf{K}_{\mathbf{f}^{(l-1)}\mathbf{Z}^{(l)}}\mathbf{K}_{\mathbf{Z}^{(l)}\mathbf{Z}^{(l)}}^{-1}\mathbf{K}_{\mathbf{Z}^{(l)}\mathbf{f}^{(l-1)}}\right). \end{aligned} \tag{4}$$

Here, we denote $\mathbf{K}_{\mathbf{ab}}^{(l)}$ as the kernel matrix at layer $l$ evaluated between sets $\mathbf{a}$ and $\mathbf{b}$. The input to the first layer is defined as $\mathbf{f}^{(0)} := \mathbf{x}$.

## 3.2 DENOISING DIFFUSION VARIATIONAL INFERENCE (DDVI)

Denoising Diffusion Variational Inference (DDVI) (Xu et al., 2024c) is a recently proposed method for performing posterior inference over inducing variables in Deep Gaussian Processes (DGPs). It draws inspiration from the success of score-based generative modeling and diffusion probabilistic models (Ho et al., 2020; Song et al., 2020), and adapts these ideas to variational inference by modeling the variational posterior as the solution to a reverse-time stochastic differential equation (SDE). Traditional variational inference in DGPs typically relies on simple, factorized Gaussian approximations over the inducing variables $\mathbf{U} = \{\mathbf{u}^{(l)}\}_{l=1}^{L}$, which are often too restrictive to capture the complex, multimodal posterior arising from deep hierarchical GPs. DDVI seeks to construct more expressive variational distributions by modeling them as the terminal distribution of a *reverse-time diffusion process*, effectively defining a flexible transformation from a known fixed initial distribution to the posterior.

**Variational posterior via reverse diffusion.** DDVI defines the variational distribution as the marginal at time $t = 1$ of a reverse-time SDE $Q_\phi(\mathbf{U}_t)$:

$$d\mathbf{U}_t = \left[ f(\mathbf{U}_t, t) - g(t)^2 \nabla_{\mathbf{U}_t} \log p_t(\mathbf{U}_t) \right] dt + g(t) \, d\mathbf{W}_t, \quad t \in [0, 1], \tag{5}$$

where $\mathbf{U}_0 \sim \mathcal{N}(\mathbf{0}, \sigma^2 \mathbf{I})$ is the fixed initial distribution, $f$ and $g$ are drift and diffusion coefficients, and $p_t(\mathbf{U}_t)$ is the marginal law at time $t$ under the reverse process. The reverse SDE is not simulated directly; instead, the score function $\nabla_{\mathbf{U}_t} \log p_t(\mathbf{U}_t)$ in the drift is parameterized by a neural network $s_\phi(\mathbf{U}_t, t)$ to approximate the optimal reverse dynamics.

**Diffusion-based ELBO.** Rather than relying on score estimation (e.g., denoising score matching), DDVI employs a variational diffusion framework to derive a tractable evidence lower bound (ELBO). Specifically, the reverse-time process $\mathbf{U}_{t\in[0,1]}$ is treated as a variational diffusion process, and the ELBO is expressed using the Girsanov formula for likelihood ratios between stochastic processes:

$$\mathcal{L}_{\text{DDVI}} = \mathbb{E}_{\mathbf{U}_{0:1} \sim Q_\phi} \left[ -\frac{1}{2\sigma^2} \|\mathbf{U}_1\|_2^2 + \log p\left(\mathbf{y} \mid \mathbf{f}^{(L)}\right) - \frac{1}{2} \int_0^1 g(t)^2 \left\| \frac{1}{\kappa_t} \mathbf{U}_t + s_\phi(t, \mathbf{U}_t) \right\|_2^2 dt \right.$$

$$\left. + \log p_{\text{prior}}(\mathbf{U}_1) - \text{KL}\left( \mathcal{N}(0, \sigma^2 \mathbf{I}) \| \mathcal{N}(0, \kappa_1 \mathbf{I}) \right) \right], \tag{6}$$

where $Q_\phi$ denotes the pathwise density of the variational reverse-time SDE, $\mathbf{U}_1$ is the terminal state of the reverse diffusion SDE using $s_\phi(\cdot, t)$, $\mathbf{f}^{(L)}$ denotes the forward inference of the DGP at $\mathbf{U}_1$.

**Training and implementation.** In practice, DDVI jointly trains the variational drift network $s_\phi$ and DGP hyperparameters $\gamma$ by maximizing $\mathcal{L}_{\text{DDVI}}$ using stochastic gradient descent. Sampling from $Q_\phi(\mathbf{U}_t)$ is achieved by solving the reverse SDE from $\mathbf{U}_0 \sim \mathcal{N}(\mathbf{0}, \sigma^2 \mathbf{I})$, which allows for reparameterized gradients through the sampled trajectories.

**Limitations.** Although DDVI offers a flexible and theoretically grounded approach for inference in deep GPs, it still seems to face two key drawbacks: first, it uses an unconditional Gaussian initialization $\mathbf{U}_0 \sim \mathcal{N}(0, \sigma^2 I)$ for the reverse diffusion, which typically appears far from the true posterior over inducing variables, so the reverse-time SDE must follow a long, complex trajectory to reach the target distribution, resulting in inefficient inference, higher variance, and slower convergence; second, this initialization is not conditioned on observations, so sampling remains input-agnostic rather than amortized and scalable. These limitations motivate our Diffusion Bridge Variational Inference (DBVI), which introduces observation-conditioned diffusion bridges and amortized initializations to enable more efficient and accurate posterior inference.

### 3.3 DBVI: OBSERVATION-CONDITIONED DIFFUSION BRIDGE

We begin by introducing the data-dependent initialization of DBVI. Instead of starting from a fixed Gaussian prior as in DDVI, we amortize the mean of the initial distribution using a neural network:

$$p_0^\theta(\mathbf{U}_0 \mid \mathbf{x}) = \mathcal{N}(\mathbf{U}_0; \mu_\theta(\mathbf{x}), \sigma^2 \mathbf{I}), \tag{7}$$

where only the mean $\mu_\theta(\mathbf{x})$ depends on the data, while the variance $\sigma^2$ is kept fixed. This initialization provides a closer match to the posterior and shortens the diffusion trajectory. To formalize the resulting dynamics, we now turn to a diffusion bridge representation.

**Proposition 1** (Forward & Reverse SDE under Doob's $h$-transform). *Let the initial constraint be encoded by the Doob $h$-transform with*

$$h(\mathbf{U}_t, t, \mathbf{U}_0) \;=\; \nabla_{\mathbf{U}_t} \log p(\mathbf{U}_0 \mid \mathbf{U}_t), \tag{8}$$

*Then the* forward bridge *has drift*

$$\tilde{f}(\mathbf{U}_t, t, \mathbf{U}_0) \;=\; f(\mathbf{U}_t, t) \;+\; g(t)^2\, h(\mathbf{U}_t, t, \mathbf{U}_0), \tag{9}$$

*with the same diffusion coefficient $g(t)$. Moreover, the* reverse-time bridge SDE *is*

$$\mathrm{d}\mathbf{U}_t \;=\; \Big[f(\mathbf{U}_t, t) \;-\; g(t)^2 s_{cond}(\mathbf{U}_t, t, \mathbf{U}_0)\Big]\mathrm{d}t \;+\; g(t)\, \mathrm{d}\mathbf{W}_t, \tag{10}$$

*Equivalently, the **conditional score** equals $s_{cond}(\mathbf{U}_t, t, \mathbf{U}_0) = s(\mathbf{U}_t, t, \mathbf{U}_0) + h(\mathbf{U}_t, t, \mathbf{U}_0)$.*

Proposition 1 states that, by introducing Doob's $h$-transform, we can reinterpret the dynamics as a bridge process, which essentially bends the diffusion toward the posterior endpoint. The forward SDE incorporates an additional drift term that nudges the path toward the target, while the reverse-time bridge SDE involves a conditional score function $s_{\mathrm{cond}}$. This result provides the mathematical foundation for DBVI, showing how conditioning on the initialization modifies both forward and reverse dynamics.

Building on this, we next leverage the bridge process trick introduced in DDVI Xu et al. (2024c) to characterize the marginal distribution of the bridge process under a linear drift. This formulation yields a tractable Gaussian form for the bridge marginal, which will be crucial for deriving the DBVI training objective.

**Proposition 2** (Marginal of Doob-augmented bridge process). *Consider the linear forward SDE with Doob bridge correction*

$$\mathrm{d}\mathbf{U}_t = \Big[-\lambda(t)\,\mathbf{U}_t + g(t)^2\, h(\mathbf{U}_t, t, \mathbf{U}_0)\Big]\mathrm{d}t + g(t)\, \mathrm{d}\mathbf{B}_t, \tag{11}$$

*where $h(\mathbf{U}_t, t, \mathbf{U}_0) = \nabla_{\mathbf{U}_t} \log p(\mathbf{U}_0 \mid \mathbf{U}_t)$ is the Doob $h$-transform. Assume isotropic initialization*

$$p_0^\theta(\mathbf{U}_0 \mid \mathbf{x}) = \mathcal{N}\big(\mu_\theta(\mathbf{x}),\, \sigma^2 \mathbf{I}\big). \tag{12}$$

*Then for each $t \in [0, 1]$, the marginal law remains Gaussian,*

$$p_t(\mathbf{U}^{Bri} \mid \mathbf{x}) = \mathcal{N}\big(\mathbf{U}^{Bri};\, \mathbf{m}_t,\, \kappa_t\, \mathbf{I}\big), \tag{13}$$

*where the mean $\mathbf{m}_t$ and variance $\kappa_t$ satisfy the coupled ODE system*

$$\frac{\mathrm{d}}{\mathrm{d}t}\, \mathbf{m}_t = -\big(\lambda(t) + c(t)\big)\, \mathbf{m}_t + c(t)\, a(t)\, \mu_\theta(\mathbf{x}), \quad \mathbf{m}_0 = \mu_\theta(\mathbf{x}), \tag{14}$$

$$\frac{\mathrm{d}}{\mathrm{d}t}\, \kappa_t = -2\big(\lambda(t) + c(t)\big)\, \kappa_t + g(t)^2 + 2\, c(t)\, a(t)\, \sigma^2, \quad \kappa_0 = \sigma^2, \tag{15}$$

*with*

$$a(t) = e^{-\Lambda(t)}, \quad \Lambda(t) = \int_0^t \lambda(s)\, \mathrm{d}s,$$

*and correction coefficient*

$$c(t) = g(t)^2\, \frac{\sigma^2\, a(t)^2}{\big(a(t)^2 \sigma^2 + q(t)\big)\, q(t)}, \qquad q(t) = a(t)^2 \int_0^t \frac{g(r)^2}{a(r)^2}\, \mathrm{d}r.$$

*In the special case $c(t) \equiv 0$, this reduces to the bridge process trick in DDVI.*

Proposition 2 provides a closed-form expression for the bridge marginal, with mean $\mathbf{m}_t$ and variance $\kappa_t$ determined by the amortized initialization $\mu_\theta(\mathbf{x})$. This Gaussian form plays a crucial role in deriving the training objective, as it enables an explicit comparison between the amortized start distribution and the bridge marginal. In particular, it provides the analytical structure needed to express the KL divergence in a tractable score–matching form, which is then incorporated into the DBVI loss. Additional derivations and detailed proofs are provided in the Appendix. Finally, we combine the above results to obtain the DBVI training loss. In particular, we express the KL divergence in a score–matching form, yielding a tractable ELBO that can be optimized efficiently.

**Proposition 3** (DBVI loss with amortized mean). *Let $p_0^\theta(\mathbf{u} \mid \mathbf{x}) = \mathcal{N}(\mu_\theta(\mathbf{x}), \sigma^2\mathbf{I})$ be the data-dependent start, and let $(\mathbf{m}_t, \kappa_t)$ be the mean/variance of the reference bridge marginal at time $t \in [0,1]$ (given by Proposition 2 via ODEs in the Doob-augmented case). Then the pathwise KL between the variational reverse bridge $Q_\phi$ and the reference bridge admits a score–matching representation. Consequently, a tractable per–mini-batch ELBO is*

$$\ell_{\mathrm{DBVI}}(\theta, \phi, \gamma) = \mathbb{E}_{\mathbf{U}_{0:1} \sim Q_\phi}\Big[ -\log p_0^\theta(\mathbf{U}_1) + \tfrac{N}{B} \log p(\mathbf{y}_I \mid \mathbf{f}^{(L)})$$
$$-\frac{1}{2}\int_0^1 g(t)^2 \left\| \tfrac{1}{\kappa_t}(\mathbf{U}_t - \mathbf{m}_t) + s_{\mathrm{cond}}(t, \mathbf{U}_t, \mathbf{U}_0)\right\|_2^2 \, \mathrm{d}t \qquad (16)$$
$$+ \log p_{\mathrm{prior}}(\mathbf{U}_1) - \mathrm{KL}\left(\mathcal{N}(\mu_\theta(\mathbf{x}), \sigma^2\mathbf{I}) \,\big\|\, \mathcal{N}(\mathbf{m}_1, \,\kappa_1\,\mathbf{I})\right)\Big],$$

*where $B$ is the batch size, $N$ is the dataset size, $s_{\mathrm{cond}} = s_\phi + h$ is the conditional score used by the reverse bridge SDE, $\mathbf{U}_1$ is its terminal state, $\mathbf{f}^{(L)}$ denotes the DGP forward mapping at $\mathbf{U}_1$, and $\gamma$ collects DGP hyperparameters. When $\mu_\theta(\mathbf{x}) = \mathbf{0}$ (and thus $\mathbf{m}_t \equiv \mathbf{0}$), the objective recovers the original DDVI loss.*

Proposition 3 shows that DBVI departs from DDVI in two essential ways: (i) the initialization is amortized via $\mu_\theta(\mathbf{x})$, which induces a time-dependent reference mean $\mathbf{m}_t$ in the bridge marginal, and (ii) the loss involves the conditional score $s_{\mathrm{cond}} = s_\phi + h$, explicitly accounting for the bridge correction. When the amortized mean collapses to zero (so that $\mathbf{m}_t \equiv \mathbf{0}$), the objective reduces exactly to the original DDVI loss, recovering DDVI as a special case. We summarize the full training procedure of DBVI for deep Gaussian processes in Algorithm 1.

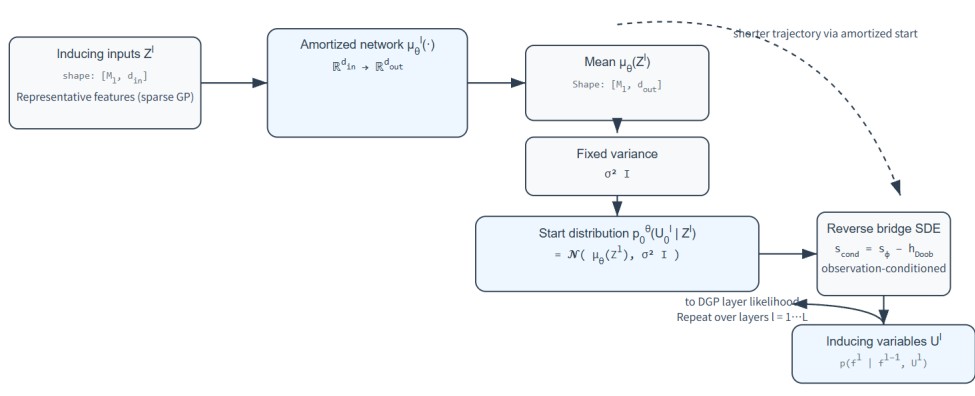

Figure 2: DBVI amortized initialization using inducing inputs $Z^{(l)}$.

## 3.4 Structure of the Amortized Network $\mu_\theta$

In DBVI, the amortized network $\mu_\theta$ provides the parameters of the initial distribution $p_0^\theta(\mathbf{U}_0 \mid \mathbf{x})$. Ideally, it would take the full dataset $\mathbf{x}$ as input and output variational parameters for the inducing variables $\mathbf{U}$. In practice, this is infeasible: full-dataset amortization is prohibitively expensive in both memory and computation, while mini-batch amortization observes only a small subset of the

---

**Algorithm 1** Diffusion Bridge Variational Inference (DBVI) for DGPs

---

**Input:** Training data $\mathbf{X}, \mathbf{y}$; mini-batch size $B$; forward drift $f$; diffusion scale $g$

**Parameters:** DGP hyperparameters $\gamma$; bridge score network $s_\phi$; amortizer $\mu_\theta(\cdot)$ with fixed variance $\sigma^2$ parameter

**Precompute reference bridge marginals** $(\mathbf{m}_t, \kappa_t)$ for $t \in [0, 1]$: numerically integrate the ODEs for $(\mathbf{m}_t, \kappa_t)$ in Prop. 2 with $a(t) = e^{-\Lambda(t)}$.

**repeat**

    Sample mini-batch indices $I \subset \{1, \ldots, N\}$ with $|I| = B$

    **Amortized start:** draw $\mathbf{U}_0 \sim p_\theta^0(\cdot \mid \mathbf{X}_I) = \mathcal{N}\big(\mu_\theta(\mathbf{X}_I), \sigma^2 \mathbf{I}\big)$

    Initialize the integral accumulator $L_0 \leftarrow 0$

    **for** $k = 0$ **to** $K - 1$ **do**

        Set $t_s \leftarrow \frac{k}{K}$,    $t_{s+1} \leftarrow \frac{k+1}{K}$,    draw $\boldsymbol{\epsilon}_{t_s} \sim \mathcal{N}(\mathbf{0}, \mathbf{I})$

        Compute conditional score: $s_{\mathrm{cond}}\big(t_s, \mathbf{U}_{t_s}, \mathbf{U}_0\big) \leftarrow s_\phi\big(t_s, \mathbf{U}_{t_s}, \mathbf{X}_I, \mathbf{y}_I\big) + h\big(t_s, \mathbf{U}_{t_s}, \mathbf{U}_0\big)$

        **Reverse bridge SDE step:**

$$\mathbf{U}_{t_{s+1}} = \mathbf{U}_{t_s} - f\big(\mathbf{U}_{t_s}, t_s\big)\Delta t + g(t_s)^2\, s_{\mathrm{cond}}\big(t_s, \mathbf{U}_{t_s}, \mathbf{U}_0\big)\Delta t + g(t_s)\sqrt{\Delta t}\,\boldsymbol{\epsilon}_{t_s}$$

        **Bridge marginal update:** Obtain $\mathbf{m}_{t_{s+1}}, \kappa_{t_{s+1}}$ and accumulate score–matching term:

$$L_{t_{s+1}} = L_{t_s} + g(t_{s+1})^2 \left\| \frac{\mathbf{U}_{t_{s+1}} - \mathbf{m}_{t_{s+1}}}{\kappa_{t_{s+1}}} + s_{\mathrm{cond}}\big(t_{s+1}, \mathbf{U}_{t_{s+1}}, \mathbf{U}_0\big) \right\|_2^2 \Delta t$$

    **end for**

    Set $\{\mathbf{u}^{(\ell)}\}_{\ell=1}^L \leftarrow \mathbf{U}_1$

    **for** $\ell = 1$ **to** $L$ **do**

        Draw $\boldsymbol{\epsilon}^{(\ell)} \sim \mathcal{N}(\mathbf{0}, \mathbf{I})$ and compute

$$\mathbf{f}^{(\ell)} = \mathbf{K}_{\mathbf{F}^{(\ell-1)}\mathbf{Z}}^{(\ell)}\big(\mathbf{K}_{\mathbf{ZZ}}^{(\ell)}\big)^{-1}\mathbf{u}^{(\ell)} + \left(\mathbf{K}_{\mathbf{F}^{(\ell-1)}\mathbf{F}^{(\ell-1)}}^{(\ell)} - \mathbf{K}_{\mathbf{F}^{(\ell-1)}\mathbf{Z}}^{(\ell)}\big(\mathbf{K}_{\mathbf{ZZ}}^{(\ell)}\big)^{-1}\mathbf{K}_{\mathbf{ZF}^{(\ell-1)}}^{(\ell)}\right)^{\frac{1}{2}}\boldsymbol{\epsilon}^{(\ell)}$$

    **end for**

    **Mini-batch ELBO:**

$$\ell_{\mathrm{DBVI}}(\theta, \phi, \gamma) = -\log p_\theta^0\big(\mathbf{u}_1 \mid \mathbf{X}_I\big) + \log p_{\mathrm{prior}}\big(\mathbf{u}_1\big) + \frac{N}{B}\log p\big(\mathbf{y}_I \mid \mathbf{F}^{(L)}\big)$$
$$- \frac{1}{2} L_1 - \mathrm{KL}\Big(\mathcal{N}\big(\mu_\theta(\mathbf{X}_I), \sigma^2 \mathbf{I}\big) \,\Big\|\, \mathcal{N}\big(\mathbf{m}_1, \kappa_1 \mathbf{I}\big)\Big)$$

    Gradient update of $\ell_{\mathrm{DBVI}}(\theta, \phi, \gamma)$

**until** $\theta, \phi, \gamma$ converge

---

data and can therefore yield biased or unstable parameter estimates. Moreover, there is a fundamental dimensional mismatch: mini-batch inputs have shape $[B, d_{\mathrm{in}}]$, whereas the inducing variables at layer $l$ lie in $[M_l, d_{\mathrm{out}}]$. Directly mapping $\mathbf{x}$ to $\mathbf{u}^{(l)}$ would require flattening high-dimensional tensors, which breaks efficient batching and scales poorly with model depth.

To avoid these issues, we use the inducing points $\mathbf{Z}^{(l)}$ as inputs to the amortizer. This follows sparse GP intuition: $\mathbf{Z}$ can be viewed as representative features of the dataset. At layer $l$, the inducing inputs $\mathbf{Z}^{(l)} \in \mathbb{R}^{M_l \times d_{\mathrm{in}}}$ are already aligned with the size of $\mathbf{u}^{(l)}$. We therefore define a layer-wise network

$$\mu_\theta^{(l)} : \mathbb{R}^{d_{\mathrm{in}}} \to \mathbb{R}^{d_{\mathrm{out}}},$$

which maps each inducing input $\mathbf{z}^{(l)}$ to a $d_{\mathrm{out}}$-dimensional representation. Applying it to all inducing points gives $\mu_\theta^{(l)}(\mathbf{Z}^{(l)}) \in \mathbb{R}^{M_l \times d_{\mathrm{out}}}$. This choice leverages information encoded in $\mathbf{Z}^{(l)}$, which is updated during training, and yields an amortization scheme whose output dimension naturally matches the inducing variables, without requiring access to the full dataset.

## 4 EXPERIMENTS

We empirically evaluate DBVI against recent state-of-the-art inference methods for Deep Gaussian Processes (DGPs). Our evaluation covers regression on UCI benchmarks, image classification on standard vision datasets, large-scale physics datasets, and an unsupervised reconstruction

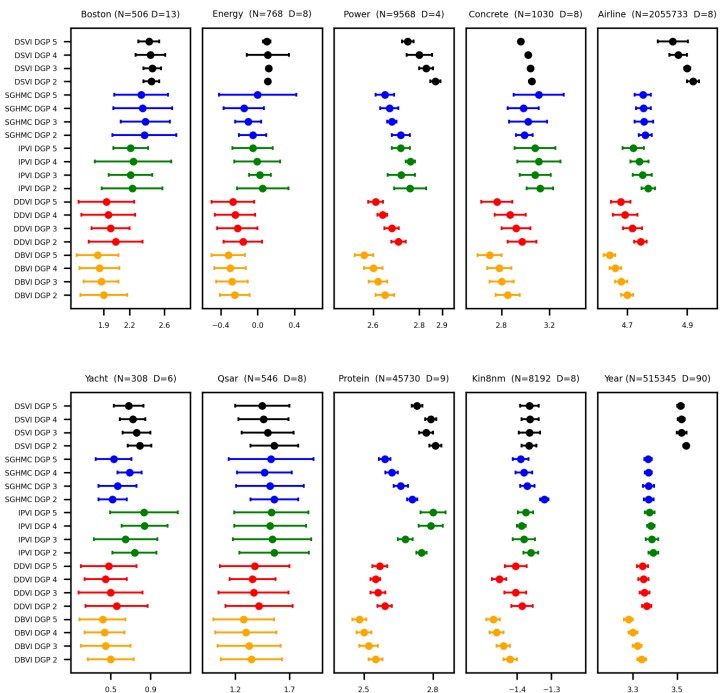

Figure 3: Test mean NLL (with one standard deviation error bars) of deep Gaussian processes with different inference methods (DDVI, IPVI, SGHMC, DSVI, and our proposed DBVI) across 10 benchmark datasets .

Table 1: Mean test accuracy (%) and training details achieved by DSVI, SGHMC, IPVI, DDVI, and our proposed DBVI on three image classification datasets. Results are shown for 3 and 4 layers as indicated, and runtime is given per iteration.

| Data Set | Model | Time3 | Iter3 | Acc3 | Time4 | Iter4 | Acc4 |
|---|---|---|---|---|---|---|---|
| MNIST | DSVI | 0.34s | 20K | 97.17 | 0.54s | 20K | 97.41 |
| | IPVI | 0.49s | 20K | 97.58 | 0.62s | 20K | 97.80 |
| | SGHMC | 1.14s | 20K | 97.25 | 1.22s | 20K | 97.55 |
| | DDVI | 0.38s | 20K | 98.84 | 0.50s | 20K | 99.01 |
| | **DBVI (ours)** | 0.41s | 20K | **99.02** | 0.55s | 20K | **99.10** |
| Fashion | DSVI | 0.34s | 20K | 87.45 | 0.50s | 20K | 87.99 |
| | IPVI | 0.48s | 20K | 88.23 | 0.61s | 20K | 88.90 |
| | SGHMC | 1.21s | 20K | 86.88 | 1.25s | 20K | 87.08 |
| | DDVI | 0.40s | 20K | 90.36 | 0.55s | 20K | 90.85 |
| | **DBVI (ours)** | 0.43s | 20K | **90.53** | 0.60s | 20K | **91.07** |
| CIFAR-10 | DSVI | 0.43s | 20K | 91.47 | 0.66s | 20K | 91.79 |
| | IPVI | 0.62s | 20K | 92.79 | 0.78s | 20K | 93.52 |
| | SGHMC | 8.04s | 20K | 92.62 | 8.61s | 20K | 92.94 |
| | DDVI | 0.45s | 20K | 95.23 | 0.69s | 20K | 95.56 |
| | **DBVI (ours)** | 0.49s | 20K | **95.42** | 0.74s | 20K | **95.68** |

task. Across these diverse settings, we assess both predictive performance and posterior quality, with particular attention to convergence behavior and scalability. The goal of our experiments is to demonstrate that DBVI consistently improves predictive accuracy and uncertainty estimation while remaining computationally efficient.

Table 2: Test AUC values for large-scale classification datasets. Uses random 90% / 10% training and test splits.

|  |  | SUSY | HIGGS |
|---|---|---|---|
|  | $N$ | 5,500,000 | 11,000,000 |
|  | $D$ | 18 | 28 |
| DSVI $M = 128$ | $L = 2$ | 0.876 | 0.830 |
|  | $L = 3$ | 0.877 | 0.837 |
|  | $L = 4$ | 0.878 | 0.841 |
|  | $L = 5$ | 0.878 | 0.846 |
| IPVI $M = 128$ | $L = 2$ | 0.879 | 0.843 |
|  | $L = 3$ | 0.882 | 0.847 |
|  | $L = 4$ | 0.883 | 0.850 |
|  | $L = 5$ | 0.883 | 0.852 |
| SGHMC $M = 128$ | $L = 2$ | 0.879 | 0.842 |
|  | $L = 3$ | 0.881 | 0.846 |
|  | $L = 4$ | 0.883 | 0.850 |
|  | $L = 5$ | 0.884 | 0.853 |
| DDVI $M = 128$ | $L = 2$ | **0.883** | **0.849** |
|  | $L = 3$ | **0.885** | **0.852** |
|  | $L = 4$ | **0.887** | **0.856** |
|  | $L = 5$ | **0.886** | **0.857** |
| DBVI (ours) $M = 128$ | $L = 2$ | **0.885** | **0.851** |
|  | $L = 3$ | **0.887** | **0.854** |
|  | $L = 4$ | **0.889** | **0.858** |
|  | $L = 5$ | **0.889** | **0.859** |

## 4.1 BASELINES AND SETUP

We compare DBVI with the following baselines: **DSVI** (Salimbeni & Deisenroth, 2017), the standard mean-field Gaussian variational approximation; **IPVI** (Yu et al., 2019), which parameterizes the posterior with a neural network trained adversarially; **SGHMC** (Havasi et al., 2018), a sampling-based inference approach; and **DDVI** (Xu et al., 2024c), the diffusion-based inference method upon which DBVI builds.

All models use RBF kernels and $M = 128$ inducing points per layer unless otherwise specified. For fairness, we adopt identical initialization and hyperparameter ranges across methods, and optimize using Adam with learning rate 0.01.

## 4.2 REGRESSION ON UCI BENCHMARKS

We evaluate our method on 10 widely used UCI regression datasets with sample sizes $N$ ranging from a few hundred to over two million, using an 80/20 train/test split. We report root mean squared error (RMSE) and negative log-likelihood (NLL) on the held-out test data, as summarized in Figure 3 and Figure 4 . Consistent with prior work, we consider deep Gaussian processes with 2–5 layers. The results demonstrate that DBVI consistently outperforms all baseline methods, with particularly pronounced gains on large-scale datasets such as YearMSD and Airline, where unconditional DDVI suffers from slow convergence. By leveraging amortized bridge initialization, DBVI effectively shortens the diffusion path length, resulting in both improved posterior approximation and lower predictive error. Figures 5, 6, and 7 show the test RMSE of a 2-layer DGP trained with DDVI and DBVI during the early stage of optimization.

Table 3: Mean RMSE and NLL achieved by DSVI, SGHMC, IPVI, DDVI, and our proposed DBVI on the GPLVM data recovery task (Frey Faces). Standard deviation is shown in parentheses. Runtime is given per iteration.

| Data Set | Model | Time | Iter | RMSE | NLL |
|---|---|---|---|---|---|
| Frey Faces | DSVI | 0.32s | 20K | 8.32 (0.20) | 1.49 (0.02) |
| | IPVI | 0.42s | 20K | 7.91 (0.40) | 1.33 (0.02) |
| | SGHMC | 1.13s | 20K | 7.95 (0.30) | 1.36 (0.03) |
| | DDVI | 0.36s | 20K | 7.64 (0.20) | 1.17 (0.01) |
| | **DBVI (ours)** | 0.40s | 20K | **7.52** (0.18) | **1.12** (0.01) |

## 4.3 IMAGE CLASSIFICATION

We next evaluate DBVI on MNIST, Fashion-MNIST, and CIFAR-10. For CIFAR-10, we adopt ResNet-20 convolutional features as inputs to the DGP classifier. We report both test accuracy and per-iteration runtime in Table 1. Across all three datasets, DBVI consistently surpasses DDVI and other baselines. In particular, on CIFAR-10 with 4-layer DGPs, DBVI achieves an accuracy of $95.68\%$, slightly higher than DDVI's $95.56\%$. These findings underscore the advantage of amortized conditioning in handling complex high-dimensional posteriors.

## 4.4 LARGE-SCALE CLASSIFICATION

We further evaluate DBVI on two large-scale physics datasets, SUSY (5.5M examples) and HIGGS (11M examples). We report AUC scores with 2–5 layer DGPs under random $90/10$ train-test splits. As shown in Table 2, DBVI consistently outperforms DDVI across all depths, yielding the best overall AUC values. Compared to SGHMC, DBVI attains comparable or higher performance while being substantially more computationally efficient, highlighting its scalability and effectiveness in modeling complex, high-dimensional posteriors at scale.

## 4.5 UNSUPERVISED RECONSTRUCTION

Finally, we evaluate posterior quality on the Frey Faces dataset using a missing-data reconstruction task. Following prior work, we randomly mask $75\%$ of pixels for a subset of training images and task the models with recovering the originals. Table 3 reports reconstruction RMSE and test log-likelihood. DBVI achieves the lowest RMSE and highest likelihood, surpassing DDVI as well as variational and sampling-based baselines. In qualitative comparisons, reconstructions generated by DBVI are visually sharper and demonstrate better calibrated uncertainty than those of competing methods.

## 5 CONCLUSION

We introduced Diffusion Bridge Variational Inference (DBVI), a principled extension of DDVI that initiates the reverse diffusion from a learnable, input-conditioned distribution. By incorporating Doob's h-transform into the variational formulation, DBVI preserves the theoretical elegance of diffusion-based inference while substantially reducing the inference gap. Our structured amortization strategy, which conditions on inducing inputs, further enables scalable and data-efficient posterior approximation in deep Gaussian processes. Empirical results across regression, classification, and image reconstruction tasks confirm that DBVI consistently improves over DDVI and other state-of-the-art variational baselines in predictive accuracy, convergence speed, and sample efficiency. These findings highlight the benefits of bridging-based inference for scalable Bayesian learning. Future directions include extending DBVI to broader probabilistic models, integrating with advanced inducing-point strategies for large-scale applications, and exploring theoretical guarantees of diffusion bridges in variational inference.

ACKNOWLEDGEMENTS

This work was supported in part by grants from National Natural Science Foundation of China (52539005), the fundamental research program of Guangdong, China (2023A1515011281), the China Scholarship Council (202306150167), Guangdong Basic and Applied Basic Research Foundation (24202107190000687), Foshan Science and Technology Research Project(2220001018608).

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

## A    PROOF OF PROPOSITIONS

**Proposition A.1** (Forward & reverse SDE under Doob's $h$-transform). *Let the initial constraint be encoded by the Doob $h$-transform with*

$$h(\mathbf{U}_t, t, \mathbf{U}_0) \;=\; \nabla_{\mathbf{U}_t} \log p(\mathbf{U}_0 \mid \mathbf{U}_t), \tag{17}$$

*Then the* forward bridge *has drift*

$$\tilde{f}(\mathbf{U}_t, t, \mathbf{U}_0) \;=\; f(\mathbf{U}_t, t) \;+\; g(t)^2\, h(\mathbf{U}_t, t, \mathbf{U}_0), \tag{18}$$

*with the same diffusion coefficient $g(t)$. Moreover, the* reverse-time bridge SDE *is*

$$d\mathbf{U}_t \;=\; \Big[ f(\mathbf{U}_t, t) \;-\; g(t)^2 s_{cond}(\mathbf{U}_t, t, \mathbf{U}_0) \Big] dt \;+\; g(t)\, d\mathbf{W}_t, \tag{19}$$

*Equivalently, the **conditional score** equals $s_{cond}(\mathbf{U}_t, t, \mathbf{U}_0) = s(\mathbf{U}_t, t, \mathbf{U}_0) + h(\mathbf{U}_t, t, \mathbf{U}_0)$.*

*Proof.* **Setup.** Consider the forward diffusion on $[0, 1]$,

$$d\mathbf{U}_t \;=\; f(\mathbf{U}_t, t)\, dt \;+\; g(t)\, d\mathbf{B}_t, \tag{20}$$

where $\mathbf{B}_t$ is a standard Brownian motion, and $g(t) > 0$ is scalar. Assume standard regularity (existence/uniqueness, non–degeneracy, smooth strictly positive densities). Let $p_t$ denote the time–$t$ marginal density of equation 20. The (backward) Kolmogorov operator is

$$(\mathcal{L}_t \varphi)(u) \;=\; f(u, t) \cdot \nabla \varphi(u) \;+\; \tfrac{g(t)^2}{2} \, \Delta \varphi(u). \tag{21}$$

**(i) Doob $h$–transform with an initial–point constraint.** Define the space–time positive function

$$H(t, u) \;:=\; p(\mathbf{U}_0 \mid \mathbf{U}_t = u). \tag{22}$$

By the Markov property and Chapman–Kolmogorov, $H$ is space–time harmonic for $\partial_t + \mathcal{L}_t$, i.e.

$$\partial_t H(t, u) \;+\; (\mathcal{L}_t H)(t, u) \;=\; 0. \tag{23}$$

The Doob $H$–transform of equation 20 is the time–inhomogeneous Markov process with generator

$$(\mathcal{L}_t^H \varphi)(u) \;=\; H(t, u)^{-1} \big( \mathcal{L}_t(H\varphi) \big)(u). \tag{24}$$

Expanding $\mathcal{L}_t(H\varphi)$ (see remarks below) and using equation 23 to cancel the $\partial_t H$ term yields, for smooth $\varphi$,

$$\mathcal{L}_t^H \varphi(u) = f(u, t) \cdot \nabla \varphi(u) + \tfrac{g(t)^2}{2} \Delta \varphi(u) + g(t)^2 \, \nabla \log H(t, u) \cdot \nabla \varphi(u). \tag{25}$$

Therefore, the transformed process is again an Itô diffusion with the *same* diffusion coefficient $g(t)$ and drift

$$\tilde{f}(u, t, \mathbf{U}_0) = f(u, t) + g(t)^2 \, \nabla_u \log H(t, u) = f(u, t) + g(t)^2 \, \nabla_u \log p(\mathbf{U}_0 \mid u). \tag{26}$$

With the proposition's definition

$$h(\mathbf{U}_t, t, \mathbf{U}_0) \;:=\; \nabla_{\mathbf{U}_t} \log p(\mathbf{U}_0 \mid \mathbf{U}_t), \tag{27}$$

we obtain the forward bridge SDE

$$d\mathbf{U}_t = \Big[ f(\mathbf{U}_t, t) + g(t)^2 \, h(\mathbf{U}_t, t, \mathbf{U}_0) \Big] dt + g(t)\, d\mathbf{B}_t, \tag{28}$$

i.e. $\tilde{f}(\mathbf{U}_t, t, \mathbf{U}_0) = f(\mathbf{U}_t, t) + g(t)^2 h(\mathbf{U}_t, t, \mathbf{U}_0)$ with unchanged diffusion $g(t)$.

**(ii) Reverse–time SDE of the initial–point bridge.** Let $\tilde{p}_t(u) := p(\mathbf{U}_t = u \mid \mathbf{U}_0)$ denote the time-$t$ conditional density of the bridge. By the Haussmann–Pardoux time–reversal formula for diffusions with isotropic diffusion matrix $a(t) = g(t)^2 I$ (so $\nabla \cdot a \equiv 0$), the time–reversed bridge $\overleftarrow{\mathbf{U}}_t = \mathbf{U}_{1-t}$ solves

$$d\overleftarrow{\mathbf{U}}_t = \Big[ \tilde{f}(\overleftarrow{\mathbf{U}}_t, 1 - t, \mathbf{U}_0) - g(1-t)^2 \nabla \log \tilde{p}_{1-t}(\overleftarrow{\mathbf{U}}_t) \Big] dt + g(1-t)\, d\overleftarrow{\mathbf{W}}_t. \tag{29}$$

Re-indexing $t \mapsto 1 - t$ and renaming the Brownian motion gives the reverse–time SDE in forward orientation:

$$d\mathbf{U}_t = \left[ \tilde{f}(\mathbf{U}_t, t, \mathbf{U}_0) - g(t)^2 \nabla \log \tilde{p}_t(\mathbf{U}_t) \right] dt + g(t)\, d\mathbf{W}_t. \tag{30}$$

Substituting $\tilde{f}(\mathbf{U}_t, t, \mathbf{U}_0) = f(\mathbf{U}_t, t) + g(t)^2 h(\mathbf{U}_t, t, \mathbf{U}_0)$ yields

$$d\mathbf{U}_t = \left[ f(\mathbf{U}_t, t) + g(t)^2 h(\mathbf{U}_t, t, \mathbf{U}_0) - g(t)^2 \nabla \log \tilde{p}_t(\mathbf{U}_t) \right] dt + g(t)\, d\mathbf{W}_t. \tag{31}$$

Define the *conditional score*

$$s_{\mathrm{cond}}(\mathbf{U}_t, t, \mathbf{U}_0) := \nabla_{\mathbf{U}_t} \log \tilde{p}_t(\mathbf{U}_t) = \nabla_{\mathbf{U}_t} \log p(\mathbf{U}_t \mid \mathbf{U}_0), \tag{32}$$

to obtain the claimed reverse–time bridge SDE:

$$d\mathbf{U}_t = \left[ f(\mathbf{U}_t, t) - g(t)^2 s_{\mathrm{cond}}(\mathbf{U}_t, t, \mathbf{U}_0) \right] dt + g(t)\, d\mathbf{W}_t. \tag{33}$$

**(iii) Conditional score identity (for the initial–point constraint).** By Bayes' rule,

$$\log p(\mathbf{U}_t \mid \mathbf{U}_0) = \log p(\mathbf{U}_0 \mid \mathbf{U}_t) + \log p_t(\mathbf{U}_t) - \log p(\mathbf{U}_0). \tag{34}$$

Taking $\nabla_{\mathbf{U}_t}$ gives

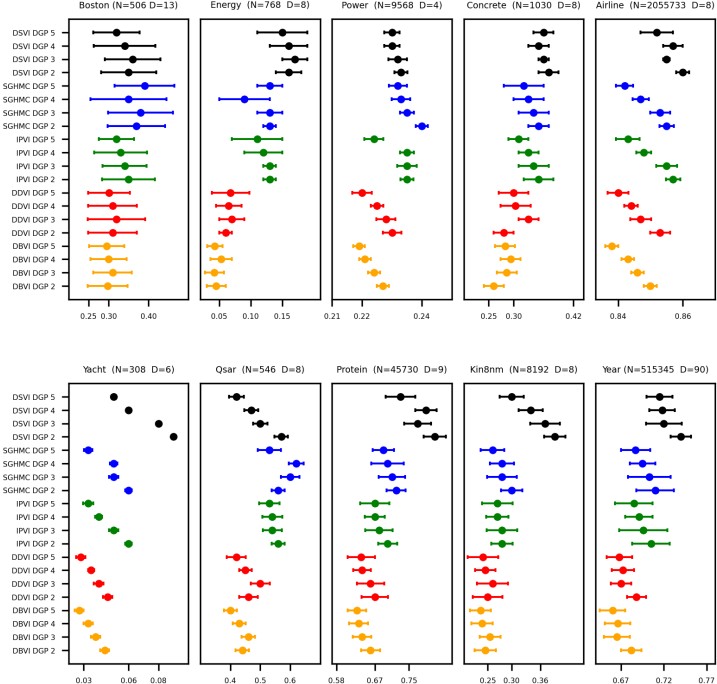

Figure 4: Test MSE (with one standard deviation error bars) of deep Gaussian processes with different inference methods (DDVI, IPVI, SGHMC, DSVI, and our proposed DBVI) across 10 benchmark datasets .

$$s_{\mathrm{cond}}(\mathbf{U}_t, t, \mathbf{U}_0) = \nabla_{\mathbf{U}_t} \log p_t(\mathbf{U}_t) + h(\mathbf{U}_t, t, \mathbf{U}_0). \tag{35}$$

If we write the unconditional score as $s(\mathbf{U}_t, t) := \nabla_{\mathbf{U}_t} \log p_t(\mathbf{U}_t)$, this is

$$s_{\mathrm{cond}} = s + h. \tag{36}$$

$\square$

**Remarks: explicit derivation for Doob $H$–transform generator.** Fix $0 \le s \le t \le 1$. The *transformed semigroup* acting on a test function $\varphi$ is

$$(T_{s,t}^H \varphi)(u) := \frac{1}{H(s,u)} \mathbb{E}\big[ H(t, \mathbf{U}_t) \, \varphi(\mathbf{U}_t) \,\big|\, \mathbf{U}_s = u \big]. \tag{37}$$

The (backward) generator $\mathcal{L}_t^H$ is defined by

$$\partial_t (T_{s,t}^H \varphi)(u) = (T_{s,t}^H \mathcal{L}_t^H \varphi)(u), \qquad \text{with} \quad T_{t,t}^H = \text{Id}. \tag{38}$$

Applying Itô to the product $H(t, \mathbf{U}_t) \, \varphi(\mathbf{U}_t)$ under the forward SDE 20 and taking conditional expectations gives

$$\frac{\mathrm{d}}{\mathrm{d}t} \mathbb{E}\big[ H(t, \mathbf{U}_t) \, \varphi(\mathbf{U}_t) \,\big|\, \mathbf{U}_s = u \big] = \mathbb{E}\Big[ \big( \partial_t + \mathcal{L}_t \big)\Big( H(t, \mathbf{U}_t) \, \varphi(\mathbf{U}_t) \Big) \,\Big|\, \mathbf{U}_s = u \Big]. \tag{39}$$

By the space–time harmonicity of $H$,

$$(\partial_t + \mathcal{L}_t) H(t, u) = 0, \tag{40}$$

we can expand $(\partial_t + \mathcal{L}_t)(H\varphi)$ and cancel the $\partial_t H$ terms cleanly. Since the backward generator is

$$(\mathcal{L}_t \psi)(u) = f(u,t) \cdot \nabla \psi(u) + \frac{g(t)^2}{2} \, \Delta \psi(u), \tag{41}$$

the *product rules* for gradient and Laplacian give

$$\nabla(H\varphi) = H \, \nabla\varphi + \varphi \, \nabla H, \qquad \Delta(H\varphi) = H \, \Delta\varphi + \varphi \, \Delta H + 2 \, \nabla H \cdot \nabla\varphi. \tag{42}$$

Hence

$$\mathcal{L}_t(H\varphi) = f \cdot \nabla(H\varphi) + \frac{g(t)^2}{2} \, \Delta(H\varphi) \tag{43}$$

$$= H \underbrace{\left( f \cdot \nabla\varphi + \frac{g(t)^2}{2} \Delta\varphi \right)}_{= \mathcal{L}_t \varphi} + \varphi \underbrace{\left( f \cdot \nabla H + \frac{g(t)^2}{2} \Delta H \right)}_{= \mathcal{L}_t H} + g(t)^2 \, \nabla H \cdot \nabla\varphi \tag{44}$$

$$= H \, \mathcal{L}_t \varphi + \varphi \, \mathcal{L}_t H + g(t)^2 \, \nabla H \cdot \nabla\varphi. \tag{45}$$

Therefore,

$$(\partial_t + \mathcal{L}_t)(H\varphi) = (\partial_t H) \, \varphi + H \, \partial_t \varphi + H \, \mathcal{L}_t \varphi + \varphi \, \mathcal{L}_t H + g(t)^2 \, \nabla H \cdot \nabla\varphi \tag{46}$$

$$= H \big( \partial_t + \mathcal{L}_t \big) \varphi + \underbrace{\big( (\partial_t + \mathcal{L}_t) H \big)}_{= 0 \text{ by } equation\ 40} \varphi + g(t)^2 \, \nabla H \cdot \nabla\varphi \tag{47}$$

$$= H \big( \partial_t + \mathcal{L}_t \big) \varphi + g(t)^2 \, \nabla H \cdot \nabla\varphi. \tag{48}$$

Dividing by $H(t, u)$ (i.e., multiplying by the scalar $H(t, u)^{-1}$) gives the pointwise identity

$$\frac{1}{H} (\partial_t + \mathcal{L}_t)(H\varphi) = (\partial_t + \mathcal{L}_t)\varphi + g(t)^2 \, \nabla\log H \cdot \nabla\varphi. \tag{49}$$

Comparing with the definition of the transformed semigroup $T_{s,t}^H$ (which removes the explicit $\partial_t$ on the left via the semigroup relation), we identify the *backward generator* of the $H$–transform at time $t$ as

$$(\mathcal{L}_t^H \varphi)(u) = (\mathcal{L}_t \varphi)(u) + g(t)^2 \, \nabla\log H(t, u) \cdot \nabla\varphi(u). \tag{50}$$

Since $H(t, u) = p(\mathbf{U}_0 \mid \mathbf{U}_t = u)$, writing $h(\mathbf{U}_t, t, \mathbf{U}_0) := \nabla_{\mathbf{U}_t} \log p(\mathbf{U}_0 \mid \mathbf{U}_t)$ yields the drift correction

$$\tilde{f}(\mathbf{U}_t, t, \mathbf{U}_0) = f(\mathbf{U}_t, t) + g(t)^2 \, h(\mathbf{U}_t, t, \mathbf{U}_0), \tag{51}$$

with the *same* diffusion coefficient $g(t)$.

**Proposition A.2** (Marginal of Doob-augmented bridge process)**.** *Consider the linear forward SDE with Doob bridge correction*

$$\mathrm{d}\mathbf{U}_t = \Big[ -\lambda(t) \, \mathbf{U}_t + g(t)^2 \, h(\mathbf{U}_t, t, \mathbf{U}_0) \Big] \mathrm{d}t + g(t) \, \mathrm{d}\mathbf{B}_t,$$

*where $h(\mathbf{U}_t, t, \mathbf{U}_0) = \nabla_{\mathbf{U}_t} \log p(\mathbf{U}_0 \mid \mathbf{U}_t)$ is the Doob h-transform. Assume isotropic initialization*

$$p_0^\theta(\mathbf{U}_0 \mid \mathbf{x}) = \mathcal{N}(\mu_\theta(\mathbf{x}), \sigma^2\mathbf{I}).$$

*Then for each $t \in [0, 1]$, the marginal law remains Gaussian,*

$$p_t(\mathbf{U}^{Bri} \mid \mathbf{x}) = \mathcal{N}(\mathbf{U}^{Bri}; \mathbf{m}_t, \kappa_t \mathbf{I}),$$

*where the mean $\mathbf{m}_t$ and variance $\kappa_t$ satisfy the coupled ODE system*

$$\frac{\mathrm{d}}{\mathrm{d}t} \mathbf{m}_t = -(\lambda(t) + c(t)) \mathbf{m}_t + c(t) a(t) \mu_\theta(\mathbf{x}), \quad \mathbf{m}_0 = \mu_\theta(\mathbf{x}), \tag{52}$$

$$\frac{\mathrm{d}}{\mathrm{d}t} \kappa_t = -2(\lambda(t) + c(t)) \kappa_t + g(t)^2 + 2 c(t) a(t) \sigma^2, \quad \kappa_0 = \sigma^2, \tag{53}$$

*with*

$$a(t) = e^{-\Lambda(t)}, \quad \Lambda(t) = \int_0^t \lambda(s) \, \mathrm{d}s,$$

*and correction coefficient*

$$c(t) = g(t)^2 \frac{\sigma^2 a(t)^2}{(a(t)^2\sigma^2 + q(t)) q(t)}, \qquad q(t) = a(t)^2 \int_0^t \frac{g(r)^2}{a(r)^2} \, \mathrm{d}r.$$

*In the special case $c(t) \equiv 0$, this reduces to the bridge process trick in DDVI..*

*Proof.* We work in the isotropic, linear-Gaussian setting adopted in the main text: the base forward SDE is

$$\mathrm{d}\mathbf{U}_t = -\lambda(t) \mathbf{U}_t \, \mathrm{d}t + g(t) \, \mathrm{d}\mathbf{B}_t, \qquad \mathbf{U}_0 \sim \mathcal{N}(\mu_\theta(\mathbf{x}), \sigma^2\mathbf{I}), \tag{54}$$

with $a(t) = e^{-\Lambda(t)}$ and $\Lambda(t) = \int_0^t \lambda(s) \, \mathrm{d}s$. In this OU setting,

$$\mathbb{E}[\mathbf{U}_t] = a(t) \mu_\theta(\mathbf{x}), \qquad \mathrm{Cov}(\mathbf{U}_t) = a(t)^2\sigma^2\mathbf{I} + q(t) \mathbf{I}, \tag{55}$$

where

$$q(t) = a(t)^2 \int_0^t \frac{g(r)^2}{a(r)^2} \, \mathrm{d}r. \tag{56}$$

Moreover, the joint law of $(\mathbf{U}_0, \mathbf{U}_t)$ is Gaussian with

$$\mathrm{Cov}(\mathbf{U}_0, \mathbf{U}_t) = \sigma^2 a(t) \mathbf{I}. \tag{57}$$

These identities are standard and match the bridge process used in the DDVI paper.

**Step 1: Conditional $p(\mathbf{U}_0 \mid \mathbf{U}_t)$.** By joint-Gaussian conditioning,

$$p(\mathbf{U}_0 \mid \mathbf{U}_t) = \mathcal{N}(\boldsymbol{m}_{0|t}, \boldsymbol{S}_{0|t}), \tag{58}$$

with isotropic (scalar) parameters

$$\boldsymbol{m}_{0|t} = \mu_\theta(\mathbf{x}) + K(t) (\mathbf{U}_t - a(t)\mu_\theta(\mathbf{x})), \qquad \boldsymbol{S}_{0|t} = S(t) \mathbf{I}, \tag{59}$$

where

$$K(t) = \frac{\sigma^2 a(t)}{a(t)^2\sigma^2 + q(t)}, \qquad S(t) = \sigma^2 - \frac{\sigma^4 a(t)^2}{a(t)^2\sigma^2 + q(t)} = \frac{\sigma^2 q(t)}{a(t)^2\sigma^2 + q(t)}. \tag{60}$$

**Step 2: Doob h-transform (bridge) term.** Define the Doob correction

$$h(\mathbf{U}_t, t, \mathbf{U}_0) = \nabla_{\mathbf{U}_t} \log p(\mathbf{U}_0 \mid \mathbf{U}_t). \tag{61}$$

Since $p(\mathbf{U}_0 \mid \mathbf{U}_t) = \mathcal{N}(\boldsymbol{m}_{0|t}, S(t)\mathbf{I})$ and $\boldsymbol{m}_{0|t}$ is affine in $\mathbf{U}_t$ with coefficient $K(t)\mathbf{I}$, we have

$$\nabla_{\mathbf{U}_t} \log p(\mathbf{U}_0 \mid \mathbf{U}_t) = (\nabla_{\mathbf{U}_t}\boldsymbol{m}_{0|t})^\top S(t)^{-1} (\mathbf{U}_0 - \boldsymbol{m}_{0|t}) = \frac{K(t)}{S(t)} (\mathbf{U}_0 - \boldsymbol{m}_{0|t}). \tag{62}$$

Hence the Doob-augmented forward SDE reads

$$\mathrm{d}\mathbf{U}_t = \left[ -\lambda(t) \mathbf{U}_t + g(t)^2 \frac{K(t)}{S(t)} (\mathbf{U}_0 - \boldsymbol{m}_{0|t}) \right]\mathrm{d}t + g(t) \, \mathrm{d}\mathbf{B}_t. \tag{$\star$}$$

**Step 3: Mean dynamics.** Let $\mathbf{m}_t := \mathbb{E}[\mathbf{U}_t]$. Taking expectations in $(\star)$ and using $\mathbb{E}[\mathbf{U}_0] = \mu_\theta(\mathbf{x})$ and $\boldsymbol{m}_{0|t} = \mu_\theta(\mathbf{x}) + K(t)\big(\mathbf{U}_t - a(t)\mu_\theta(\mathbf{x})\big)$ gives

$$\mathbb{E}\big[\mathbf{U}_0 - \boldsymbol{m}_{0|t}\big] = \mu_\theta(\mathbf{x}) - \Big(\mu_\theta(\mathbf{x}) + K(t)\big(\mathbf{m}_t - a(t)\mu_\theta(\mathbf{x})\big)\Big) = K(t)\big(a(t)\mu_\theta(\mathbf{x}) - \mathbf{m}_t\big). \quad (63)$$

Therefore,

$$\frac{\mathrm{d}}{\mathrm{d}t}\,\mathbf{m}_t = -\lambda(t)\,\mathbf{m}_t \;+\; g(t)^2\,\frac{K(t)}{S(t)}\,K(t)\,\big(a(t)\mu_\theta(\mathbf{x}) - \mathbf{m}_t\big). \quad (64)$$

Introduce the scalar

$$c(t) \;=\; g(t)^2\,\frac{K(t)^2}{S(t)} = g(t)^2\,\frac{\sigma^2 a(t)^2}{\big(a(t)^2\sigma^2 + q(t)\big)\,q(t)}, \quad (65)$$

which yields

$$\frac{\mathrm{d}}{\mathrm{d}t}\,\mathbf{m}_t \;=\; -\big(\lambda(t) + c(t)\big)\,\mathbf{m}_t \;+\; c(t)\,a(t)\,\mu_\theta(\mathbf{x}), \qquad \mathbf{m}_0 = \mu_\theta(\mathbf{x}). \quad (66)$$

**Step 4: Variance dynamics.** Let $\kappa_t$ denote the (isotropic) variance so that $\mathrm{Cov}(\mathbf{U}_t) = \kappa_t \mathbf{I}$. From Itô's lemma for $\mathbf{U}_t\mathbf{U}_t^\top$ under $(\star)$ and isotropy,

$$\frac{\mathrm{d}}{\mathrm{d}t}\,\kappa_t = -2\lambda(t)\,\kappa_t \;+\; g(t)^2 \;+\; 2\,g(t)^2\,\frac{K(t)}{S(t)}\,\mathrm{Cov}\big(\mathbf{U}_t,\,\mathbf{U}_0 - \boldsymbol{m}_{0|t}\big)_{\mathrm{scalar}}. \quad (67)$$

Using $\mathrm{Cov}(\mathbf{U}_t, \mathbf{U}_0) = a(t)\sigma^2\mathbf{I}$ and $\boldsymbol{m}_{0|t} = \mu_\theta(\mathbf{x}) + K(t)\big(\mathbf{U}_t - a(t)\mu_\theta(\mathbf{x})\big)$, we get

$$\mathrm{Cov}\big(\mathbf{U}_t, \boldsymbol{m}_{0|t}\big) = K(t)\,\mathrm{Cov}\big(\mathbf{U}_t, \mathbf{U}_t\big) = K(t)\,\kappa_t\,\mathbf{I}, \quad (68)$$

hence

$$\mathrm{Cov}\big(\mathbf{U}_t,\,\mathbf{U}_0 - \boldsymbol{m}_{0|t}\big) = a(t)\sigma^2\,\mathbf{I} - K(t)\kappa_t\,\mathbf{I}. \quad (69)$$

Plugging in and using $c(t) = g(t)^2 K(t)^2/S(t)$,

$$\frac{\mathrm{d}}{\mathrm{d}t}\,\kappa_t = -2\lambda(t)\,\kappa_t + g(t)^2 + 2\,\frac{c(t)}{K(t)}\Big(a(t)\sigma^2 - K(t)\kappa_t\Big) = -2\big(\lambda(t) + c(t)\big)\kappa_t + g(t)^2 + 2\,c(t)\,a(t)\,\sigma^2, \quad (70)$$

with $\kappa_0 = \sigma^2$.

**Step 5: Gaussian form.** The drift in $(\star)$ is affine in $(\mathbf{U}_t, \mathbf{U}_0)$, and the driving noise is Gaussian. Therefore $(\mathbf{U}_t, \mathbf{U}_0)$ remains jointly Gaussian, and the marginal $p_t(\mathbf{U}^{\mathrm{Bri}} \mid \mathbf{x})$ is Gaussian with mean $\mathbf{m}_t$ and variance $\kappa_t\mathbf{I}$. This proves the claimed form and the coupled ODE system. In the special case $c(t) \equiv 0$ (no Doob correction), the system reduces to the closed-form bridge process in DDVI paper, i.e., $\mathbf{m}_t = a(t)\mu_\theta(\mathbf{x})$ and $\kappa_t = a(t)^2\sigma^2 + a(t)^2\int_0^t \frac{g(r)^2}{a(r)^2}\mathrm{d}r$. $\qquad\square$

**Proposition A.3** (DBVI loss with amortized mean). *Let $p_0^\theta(\mathbf{u} \mid \mathbf{x}) = \mathcal{N}(\mu_\theta(\mathbf{x}), \sigma^2\mathbf{I})$ be the data-dependent start, and let $(\mathbf{m}_t, \kappa_t)$ be the mean/variance of the reference bridge marginal at time $t \in [0, 1]$ (given by Proposition 2 via ODEs in the Doob-augmented case). Then the pathwise KL between the variational reverse bridge $Q_\phi$ and the reference bridge admits a score–matching representation. Consequently, a tractable per–mini-batch ELBO is*

$$\ell_{\mathrm{DBVI}}(\theta, \phi, \gamma) = \mathbb{E}_{\mathbf{U}_{0:1}\sim Q_\phi}\Big[ -\log p_0^\theta(\mathbf{U}_1) + \frac{N}{B}\log p\big(\mathbf{y}_I \mid \mathbf{f}^{(L)}\big)$$
$$-\frac{1}{2}\int_0^1 g(t)^2\left\|\frac{\mathbf{U}_t - \mathbf{m}_t}{\kappa_t} + s_{\mathrm{cond}}\big(t, \mathbf{U}_t, \mathbf{U}_0\big)\right\|_2^2\mathrm{d}t$$
$$-\mathrm{KL}\big(\mathcal{N}(\mu_\theta(\mathbf{x}), \sigma^2\mathbf{I}) \,\big\|\, \mathcal{N}(\mathbf{m}_1,\, \kappa_1\,\mathbf{I})\big) + \log p_{\mathrm{prior}}(\mathbf{U}_1)\Big], \quad (71)$$

*where $B$ is the batch size, $N$ is the dataset size, $s_{\mathrm{cond}} = s_\phi + h$ is the conditional score used by the reverse bridge SDE, $\mathbf{U}_1$ is its terminal state, $\mathbf{f}^{(L)}$ denotes the DGP forward mapping at $\mathbf{U}_1$, and $\gamma$ collects DGP hyperparameters. When $\mu_\theta(\mathbf{x}) = \mathbf{0}$ (and thus $\mathbf{m}_t \equiv \mathbf{0}$), the objective recovers the original DDVI loss.*

*Proof.* **Setup.** Let $Q_\phi$ be the path law of the reverse-time bridge SDE

$$d\mathbf{U}_t = \left[f(\mathbf{U}_t, t) - g(t)^2 s_{\text{cond}}(t, \mathbf{U}_t, \mathbf{U}_0)\right]dt + g(t)\,d\mathbf{W}_t, \qquad s_{\text{cond}} := s_\phi + h,$$

with $h(\mathbf{U}_t, t, \mathbf{U}_0) = \nabla_{\mathbf{U}_t} \log p(\mathbf{U}_0 \mid \mathbf{U}_t)$. Let $P$ be the forward reference diffusion and $P^{\text{Bri}}$ be the auxiliary forward process that shares the same drift/diffusion as $P$ but starts from $p_0^\theta(\mathbf{u} \mid \mathbf{x}) = \mathcal{N}(\mu_\theta(\mathbf{x}), \sigma^2 \mathbf{I})$. Assume Novikov's condition so that Girsanov applies.

**(A) Path term $\Rightarrow$ score–matching quadratic.** By reverse-time Girsanov (see lemma 1), the process KL splits into a path term plus a boundary term; the path term is

$$\mathbb{E}_{\mathbf{U}_{0:1} \sim Q_\phi}\left[\log \frac{dQ_\phi}{dP^{\text{Bri}}}\right] = \frac{1}{2} \int_0^1 g(t)^2\, \mathbb{E}_{Q_\phi}\left[\left\|\, s_{\text{cond}}(t, \mathbf{U}_t, \mathbf{U}_0) - s_{\text{Bri}}(t, \mathbf{U}_t)\,\right\|_2^2\right]dt, \qquad (72)$$

where $s_{\text{Bri}}(t, \mathbf{U}_t) := \nabla_{\mathbf{U}_t} \log p_t(\mathbf{U}_t)$ is the (marginal) score of the reference bridge at time $t$. By Proposition 2, the marginal is Gaussian $p_t(\mathbf{U}_t) = \mathcal{N}(\mathbf{m}_t, \kappa_t \mathbf{I})$, hence

$$s_{\text{Bri}}(t, \mathbf{U}_t) = -\frac{\mathbf{U}_t - \mathbf{m}_t}{\kappa_t}. \qquad (73)$$

Substituting this into equation 72 yields the integrand

$$\left\|\, s_{\text{cond}} - s_{\text{Bri}}\,\right\|_2^2 = \left\|\frac{\mathbf{U}_t - \mathbf{m}_t}{\kappa_t} + s_{\text{cond}}(t, \mathbf{U}_t, \mathbf{U}_0)\right\|_2^2, \qquad (74)$$

which is exactly the score–matching form stated in the proposition.

**(B) Boundary term.** Since $P$ and $P^{\text{Bri}}$ share dynamics and only differ at the start, the remaining term in the KL decomposition collapses to a boundary difference (as in DDVI):

$$\mathbb{E}_{Q_\phi}\left[\log \frac{dP^{\text{Bri}}}{dP}\right] = \mathbb{E}_{Q_\phi}\left[\log \frac{p_0^\theta(\mathbf{U}_1)}{p_1^{\text{Bri}}(\mathbf{U}_1)}\right] = -\text{KL}\left(\mathcal{N}(\mu_\theta(\mathbf{x}), \sigma^2 \mathbf{I}) \,\big\|\, \mathcal{N}(\mathbf{m}_1, \kappa_1 \mathbf{I})\right), \qquad (75)$$

where $p_1^{\text{Bri}} = \mathcal{N}(\mathbf{m}_1, \kappa_1 \mathbf{I})$ follows from Proposition 2.

**(C) Collecting the ELBO terms.** Adding the model likelihood and prior contributions (as in DDVI; mini-batch scaled by $N/B$) yields, up to constants independent of $(\theta, \phi, \gamma)$,

$$\begin{aligned}
\ell_{\text{DBVI}}(\theta, \phi, \gamma) = \mathbb{E}_{\mathbf{U}_{0:1} \sim Q_\phi}\Big[ &- \log p_0^\theta(\mathbf{U}_1) + \frac{N}{B} \log p(\mathbf{y}_I \mid \mathbf{f}^{(L)}) \\
&- \frac{1}{2} \int_0^1 g(t)^2 \left\|\frac{\mathbf{U}_t - \mathbf{m}_t}{\kappa_t} + s_{\text{cond}}(t, \mathbf{U}_t, \mathbf{U}_0)\right\|_2^2 dt \\
&- \text{KL}\left(\mathcal{N}(\mu_\theta(\mathbf{x}), \sigma^2 \mathbf{I}) \,\big\|\, \mathcal{N}(\mathbf{m}_1, \kappa_1 \mathbf{I})\right) + \log p_{\text{prior}}(\mathbf{U}_1)\Big],
\end{aligned} \qquad (76)$$

which matches the statement.

**(D) Reduction to DDVI.** If $\mu_\theta(\mathbf{x}) \equiv \mathbf{0}$, then $\mathbf{m}_t \equiv \mathbf{0}$ and the loss reduces to the original DDVI objective (same path integral with $s_{\text{cond}} = s_\phi + h$ and the usual boundary terms). $\square$

**Lemma 1** (Reverse-time Girsanov: path term as score–matching)**.** *Fix $T = 1$. Consider two reverse-time SDEs on $\mathbb{R}^d$ with the same diffusion scale $g(t) > 0$:*

$$(Q) \quad d\mathbf{U}_t = \left(f(\mathbf{U}_t, t) - g(t)^2 s_{\text{cond}}(t, \mathbf{U}_t, \mathbf{U}_0)\right)dt + g(t)\,d\mathbf{W}_t^{(Q)}, \qquad (77)$$

$$(P^{\text{Bri}}) \quad d\mathbf{U}_t = \left(f(\mathbf{U}_t, t) - g(t)^2 s_{\text{Bri}}(t, \mathbf{U}_t)\right)dt + g(t)\,d\mathbf{W}_t^{(P)}, \qquad (78)$$

*where $s_{\text{cond}} = s_\phi + h$ is the conditional (bridge-corrected) score, and $s_{\text{Bri}}(t, \mathbf{u}) = \nabla_{\mathbf{u}} \log p_t(\mathbf{u})$ is the marginal score of the reference bridge (with marginal $p_t(\mathbf{u}) = \mathcal{N}(\mathbf{m}_t, \kappa_t \mathbf{I})$). Assume standard*

*integrability (e.g. Novikov) so that Girsanov applies and both path laws $Q_\phi$ and $P^{\mathrm{Bri}}$ are mutually absolutely continuous on $\mathcal{F}_T$. Then the pathwise Kullback–Leibler divergence decomposes as*

$$\mathrm{KL}(Q_\phi \,\|\, P^{\mathrm{Bri}}) \;=\; \underbrace{\tfrac{1}{2} \int_0^T g(t)^2 \, \mathbb{E}_{Q_\phi}\Big[\big\|s_{\mathrm{cond}}(t, \mathbf{U}_t, \mathbf{U}_0) - s_{\mathrm{Bri}}(t, \mathbf{U}_t)\big\|_2^2\Big] \, \mathrm{d}t}_{\textit{path term}} \;+\; \underbrace{\mathbb{E}_{Q_\phi}\left[\log \frac{q_T(\mathbf{U}_T)}{p_T(\mathbf{U}_T)}\right]}_{\textit{boundary term}},$$
(79)

*where $q_T$ and $p_T$ are the terminal densities of the two reverse processes (equivalently: the initial densities of the corresponding forward processes). In particular, the path term equals*

$$\frac{1}{2} \int_0^1 g(t)^2 \, \mathbb{E}_{Q_\phi}\big[\|\, s_{\mathrm{cond}}(t, \mathbf{U}_t, \mathbf{U}_0) - s_{\mathrm{Bri}}(t, \mathbf{U}_t)\,\|_2^2\big] \, \mathrm{d}t,$$
(80)

*which is the score–matching quadratic used in equation 72.*

*Proof.* We write both SDEs on a common probability space up to an equivalent change of measure. Let the *reference* path law be $P^{\mathrm{Bri}}$ (drift $b_P := f - g^2 s_{\mathrm{Bri}}$). Under $P^{\mathrm{Bri}}$, define the progressively measurable process

$$\vartheta_t \;:=\; \frac{b_Q(\mathbf{U}_t, t) - b_P(\mathbf{U}_t, t)}{g(t)} \;=\; -g(t)\big(s_{\mathrm{cond}}(t, \mathbf{U}_t, \mathbf{U}_0) - s_{\mathrm{Bri}}(t, \mathbf{U}_t)\big),$$
(81)

where $b_Q := f - g^2 s_{\mathrm{cond}}$. By Novikov's condition, the Dôleans–Dade exponential

$$Z_t \;=\; \exp\Big( \int_0^t \vartheta_s^\top \, \mathrm{d}\mathbf{W}_s^{(P)} - \tfrac{1}{2} \int_0^t \|\vartheta_s\|^2 \, \mathrm{d}s \Big)$$
(82)

is a true $P^{\mathrm{Bri}}$-martingale. Define a new measure $Q_\phi$ on $\mathcal{F}_T$ by $\frac{\mathrm{d}Q_\phi}{\mathrm{d}P^{\mathrm{Bri}}}\big|_{\mathcal{F}_T} = Z_T \cdot \frac{q_T(\mathbf{U}_T)}{p_T(\mathbf{U}_T)}$, i.e. we also correct the endpoint density (boundary term) so that the terminal marginal under $Q_\phi$ is $q_T$. Girsanov's theorem yields that under $Q_\phi$,

$$\mathbf{W}_t^{(Q)} \;:=\; \mathbf{W}_t^{(P)} - \int_0^t \vartheta_s \, \mathrm{d}s$$
(83)

is a Brownian motion and the drift becomes $b_Q = f - g^2 s_{\mathrm{cond}}$, i.e. the reverse SDE for $Q_\phi$.

Now compute the log Radon–Nikodym derivative and take $Q_\phi$-expectation:

$$\mathrm{KL}(Q_\phi \,\|\, P^{\mathrm{Bri}}) = \mathbb{E}_{Q_\phi}\left[\log \frac{\mathrm{d}Q_\phi}{\mathrm{d}P^{\mathrm{Bri}}}\right]$$
(84)

$$= \mathbb{E}_{Q_\phi}\left[\int_0^T \vartheta_t^\top \, \mathrm{d}\mathbf{W}_t^{(P)} - \frac{1}{2} \int_0^T \|\vartheta_t\|^2 \, \mathrm{d}t\right] \;+\; \mathbb{E}_{Q_\phi}\left[\log \frac{q_T(\mathbf{U}_T)}{p_T(\mathbf{U}_T)}\right].$$
(85)

Use $\mathrm{d}\mathbf{W}_t^{(P)} = \mathrm{d}\mathbf{W}_t^{(Q)} + \vartheta_t \, \mathrm{d}t$ to rewrite the stochastic integral:

$$\int_0^T \vartheta_t^\top \, \mathrm{d}\mathbf{W}_t^{(P)} \;=\; \int_0^T \vartheta_t^\top \, \mathrm{d}\mathbf{W}_t^{(Q)} + \int_0^T \|\vartheta_t\|^2 \, \mathrm{d}t.$$
(86)

Taking $Q_\phi$-expectation annihilates the martingale term, hence

$$\mathrm{KL}(Q_\phi \,\|\, P^{\mathrm{Bri}}) \;=\; \frac{1}{2} \mathbb{E}_{Q_\phi}\left[\int_0^T \|\vartheta_t\|^2 \, \mathrm{d}t\right] \;+\; \mathbb{E}_{Q_\phi}\left[\log \frac{q_T(\mathbf{U}_T)}{p_T(\mathbf{U}_T)}\right].$$
(87)

Finally substitute $\vartheta_t = -g(t)\big(s_{\mathrm{cond}}(t, \mathbf{U}_t, \mathbf{U}_0) - s_{\mathrm{Bri}}(t, \mathbf{U}_t)\big)$ to get

$$\frac{1}{2} \int_0^T g(t)^2 \, \mathbb{E}_{Q_\phi}\Big[\big\|s_{\mathrm{cond}}(t, \mathbf{U}_t, \mathbf{U}_0) - s_{\mathrm{Bri}}(t, \mathbf{U}_t)\big\|_2^2\Big] \, \mathrm{d}t$$
(88)

as the path term, plus the boundary correction. $\qquad\square$

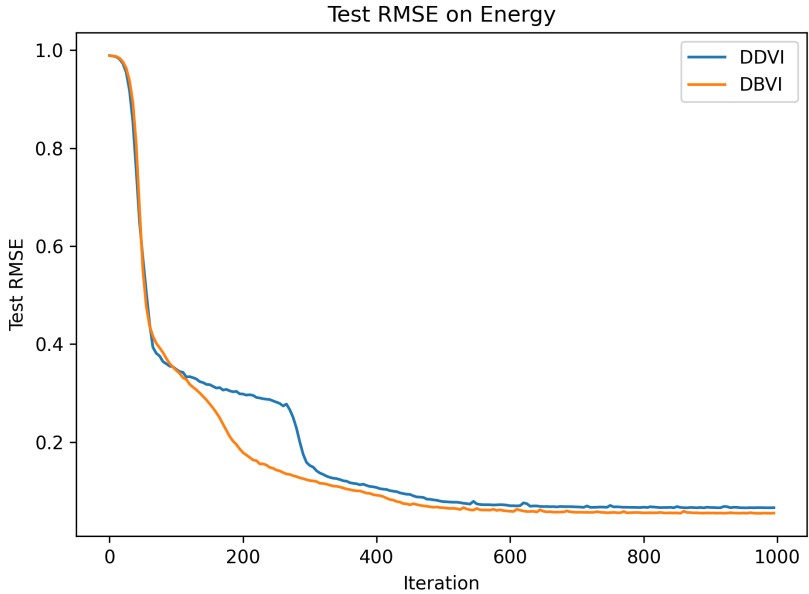

Figure 5: Comparison of DDVI and DBVI on the ENERGY dataset (Test RMSE).

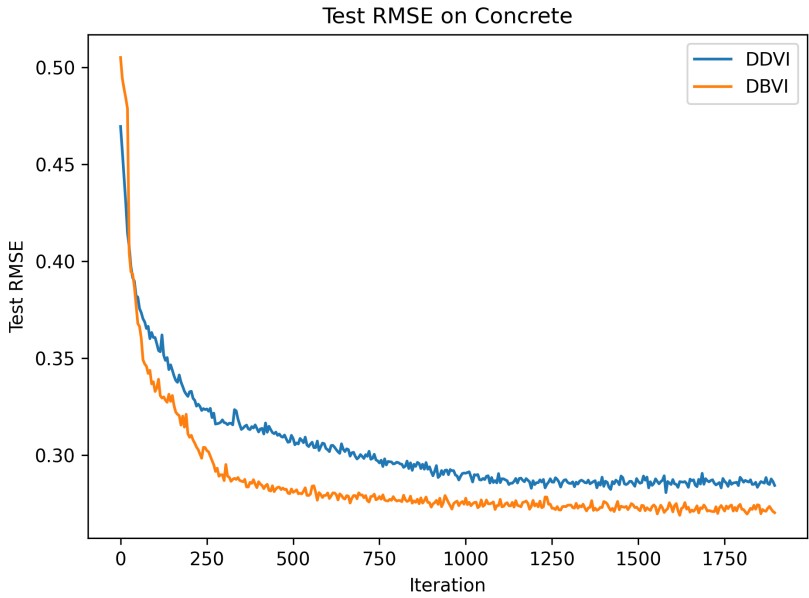

Figure 6: Comparison of DDVI and DBVI on the CONCRETE dataset (Test RMSE).

# B ABLATION STUDY

## B.1 CONVERGENCE SPEED UNDER MATCHED COMPUTE

We directly compare DDVI and DBVI under matched compute (same $T$, optimizer, and hardware) and track Test RMSE during early optimization. As shown in Figures 5, 6 and 7, DBVI reduces Test RMSE more rapidly and consistently achieves lower error than DDVI. This confirms that conditioning the reverse diffusion on a data-dependent initialization shortens the inference trajectory and improves convergence speed on the ENERGY, CONCRETE and POWER datasets.

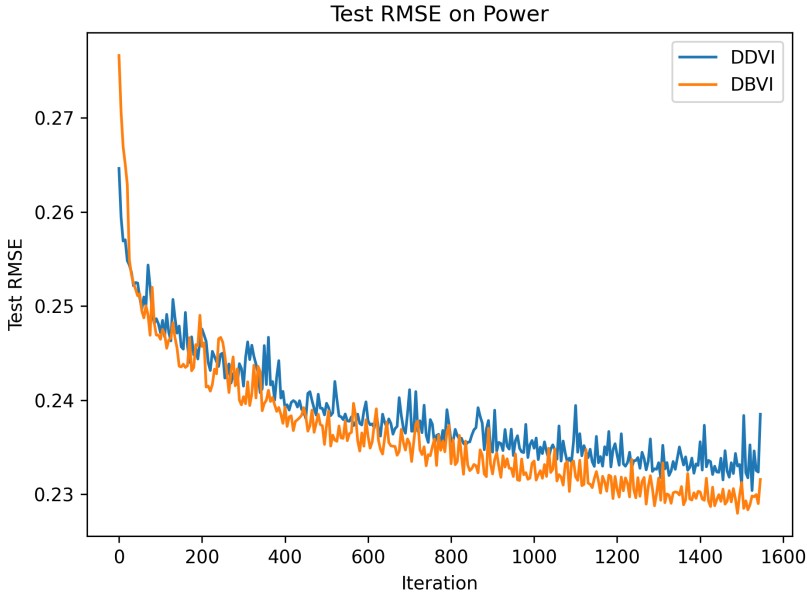

Figure 7: Comparison of DDVI and DBVI on the POWER dataset (Test RMSE).

## B.2 TRAJECTORY SHORTENING DIAGNOSTIC VIA PATH LENGTH

A key motivation of DBVI is that the reverse diffusion should traverse a shorter and less complex trajectory, since the amortized initialization is closer to the true posterior. To quantitatively diagnose this effect, we report a discrete path-length proxy for the reverse diffusion trajectory,

$$L \; = \; \mathbb{E}\left[\sum_{k=0}^{K-1}\left\|U_{t_{k+1}} - U_{t_k}\right\|_2^2\right],\tag{89}$$

where the expectation is taken over the stochasticity of the reverse SDE (and mini-batch sampling) and $\{t_k\}_{k=0}^{K}$ denotes the discretization grid of the reverse-time SDE with $K$ steps. Intuitively, $L$ measures how far the trajectory moves in latent space during sampling, and smaller values indicate a shorter reverse diffusion path.

Empirically, DBVI consistently yields substantially smaller path length than DDVI. On the Concrete dataset, DBVI achieves an average path length of $0.368$, compared to $12.167$ for DDVI. On the Energy dataset, DBVI similarly yields $0.367$ versus $11.953$ for DDVI. These results provide direct quantitative evidence that the proposed bridge initialization significantly shortens the reverse diffusion trajectory.

## C STATEMENT ON THE USE OF LARGE LANGUAGE MODELS

Large language models (LLMs) were used solely for polishing and editing the text of this manuscript.

