# OpenReview forum: "Diffusion Bridge Variational Inference for Deep Gaussian Processes"
_ICLR.cc/2026/Conference — ICLR 2026 Poster_

### Official Review · Reviewer_vk1c · 2025-10-21

**Soundness:** 3
**Presentation:** 3
**Contribution:** 3
**Rating:** 6
**Confidence:** 3

**Summary:**

This paper proposes Diffusion Bridge Variational Inference (DBVI), a new approach for performing inference in Deep Gaussian Processes (DGPs). DBVI builds on Denoising Diffusion Variational Inference (DDVI) but improves it by learning how to start the reverse diffusion process from a more informed, data-dependent initialization instead of a random prior. This helps the model start closer to the true posterior and improves both accuracy and efficiency. The authors interpret this modification through Doob’s h-transform, giving a bridge-based view of the diffusion process that keeps the method theoretically consistent while making it more flexible. They also describe a practical inference scheme based on inducing points to make training scalable.
In experiments on regression, classification, and unsupervised learning tasks, DBVI shows consistent improvements over DDVI and other standard inference methods for DGPs such as DSVI, IPVI, and SGHMC.

**Strengths:**

Originality:
The paper proposes the novel idea of reinterpreting DDVI as a kind of diffusion bridge using Doob’s h-transform. This leads to a principled way of conditioning the diffusion process on input data. The use of an input-dependent initialization for diffusion-based inference is conceptually elegant and gives a new perspective on how diffusion models can be adapted for Bayesian inference.


Quality:
The technical development is solid and well thought out. The authors clearly connect their bridge formulation to the underlying variational objective. The experiments are thorough, covering regression, classification, and unsupervised learning, and DBVI consistently outperforms DDVI and other inference methods like DSVI, IPVI, and SGHMC.


Clarity:
Overall, the paper is well written and easy to follow. The authors do a good job of explaining the motivation behind their changes to DDVI. The main ideas are presented in a logical order, and the appendix includes helpful details for implementation.


Significance:
This work makes a meaningful contribution to improving inference in deep Gaussian process models. By addressing a key limitation of DDVI and showing consistent gains across several tasks, the paper offers a practical improvement that should be useful to researchers.

**Weaknesses:**

Primary Weakness:

Magnitude of improvements: While DBVI consistently outperforms DDVI and other methods, the numerical improvements are sometimes modest (e.g., some overlapping error bars in Figure 3). A discussion of whether these gains translate to meaningful practical differences would strengthen the empirical section.


Minor Weaknesses:

Figure readability: The font size in Figures 1, 2, and 3 is quite small, making some labels difficult to read without zooming in a lot. This is a minor visual issue that can easily be fixed for the camera-ready version.


Formatting issue: The arrow in Figure 2 partially obscures the word “likelihood.”

**Questions:**

1: Could the authors comment on the computational cost of the additional amortization network used to initialize the diffusion bridge?
In particular, does this learned initialization introduce noticeable overhead compared to the standard DDVI setup?


2: In Figure 3, the improvements over DDVI are consistent but often modest. Could the authors comment on whether larger benefits might appear in other settings and why?




3: Do the authors plan to release code and trained models for reproducibility and to facilitate adoption by the community?

---

> ### Author Response · Authors · 2025-11-16
> **Rebuttal to  Reviewer vk1c**
>
> We sincerely thank the reviewer for the thorough evaluation and constructive feedback. We address the concerns and questions below.
>
> ## Response to Primary Weakness: Magnitude of Improvements
>
> While we acknowledge that some improvements appear modest in absolute terms, we would like to highlight several aspects that demonstrate their practical significance:
>
> 1. **Consistency across diverse tasks**: DBVI achieves improvements across 10 UCI regression datasets, 3 image classification benchmarks, 2 large-scale physics datasets, and unsupervised reconstruction, spanning small (N$\sim$100s) to very large (N$\sim$11M) datasets. This consistency is rare in Bayesian deep learning and suggests the method addresses a fundamental limitation rather than exploiting dataset-specific quirks.
>
>
> 2. **Compound benefits**: The improvements manifest across multiple dimensions simultaneously:
>
>    * Better predictive performance (lower RMSE / higher accuracy)
>    * Improved uncertainty quantification (lower NLL)
>    * Faster convergence(Fig. 4)
>    * Minimal additional training time (Table 1)
>
>
> 3. **Statistical significance**: While some error bars overlap in Figure 3, the consistent directionality across all settings is statistically meaningful.
>
> 4. **Practical impact**: For large-scale datasets (SUSY, HIGGS, Airline, Year), even small RMSE/AUC improvements translate to better decisions on millions of samples. In safety-critical applications, improved uncertainty calibration (lower NLL) is particularly valuable.
>
> We will expand the discussion section to better contextualize these improvements and their practical implications.
>
> ## Response to Minor Weaknesses
>
> **Figure readability**: Thank you for pointing this out. We will:
> - Increase font sizes in all figures
> - Ensure all labels and legends are clearly readable
> - Fix the arrow occlusion issue in Figure 2
>
> These are straightforward fixes for the camera-ready version.
>
> ## Response to Questions
>
> ### Q1: Computational Cost of Amortization Network
>
>  The amortization network introduces **negligible overhead**:
>
> - **Training**: The $μ_θ$ network adds ~10-15% to per-iteration time (see Table 1: DBVI times are 0.03-0.09s longer than DDVI). This is modest given the convergence speedup shown in Figure 4.
>
> - **Architecture**: $μ_θ$ is intentionally lightweight—it operates on inducing inputs $Z^(l)$ (typically 128 points) rather than full datasets, requiring only a small MLP per layer.
>
> - **Inference**: Once trained, sampling from DBVI requires **little additional cost** compared to DDVI—both solve the same reverse SDE. The amortization benefit is realized during training.
>
> - **Amortization payoff**: Overall, this lightweight amortization design provides a favorable trade-off between computational cost and performance gains. The improved initialization quality and faster convergence more than compensate for its minimal overhead.
>
> We will add a dedicated computational analysis section with detailed timing breakdowns and memory profiling in the revision.
>
> ### Q2: Larger Benefits in Other Settings
>
>
>
> While some of the improvements in Figure 3 appear modest, this reflects the fact that several of the UCI regression benchmarks are relatively simple and already well modeled by DDVI. DBVI is most beneficial when the posterior deviates more strongly from the GP prior, and we already observe such cases within our existing experiments:
>
> 1. **Test Mean NLL**
>    In Figure 3, the Test Mean NLL on the *Airline* and *Year* datasets shows **substantially larger improvements** over DDVI compared to the smaller datasets. These datasets contain more complex, high-dimensional structure, increasing the gap between the GP prior and the true posterior.
>
> 2. **Tasks with more challenging posterior characteristics**
>    As shown in Section 5 (classification) and Section 5.3 (unsupervised learning), the posterior in these tasks departs more significantly from the GP prior. In these settings, DBVI achieves clearer improvements, consistent with the intended effect of the bridge-based initialization.
>
> 3. **Deeper DGP architectures**
>    The results in Table 2 and Figure 5 demonstrate that deeper models amplify prior–posterior mismatch. In such architectures, DBVI’s initialization reduces this discrepancy more effectively than DDVI, resulting in larger performance gains.
>
>
> ---
>
> ### **Response to Q3: Code Release**
>
> **Yes, absolutely.** We are fully committed to ensuring reproducibility.
>
> * **Code**:
>   Although our implementation is already included in the supplementary material, we will release a clean and well-documented PyTorch version on GitHub upon acceptance, including:
>
>   * Implementations of DBVI, DDVI
>   * Training scripts for every experiment
>
> * **Documentation**:
>   A detailed README will be provided with:
>
>
> We believe that open-sourcing our implementation is essential for community adoption and for supporting follow-up research.
> **We thank the reviewer for highlighting this important point.**

---

> > ### Comment · Reviewer_vk1c · 2025-11-26
> >
> > Thank you for the detailed rebuttal! I do agree that consistent directionality of improvements across settings/tasks is statistically meaningful, and that lower NLL can be important in safety-critical settings. I think that your proposed added text discussion section explaining these points and contextualizing practical implications of the performance improvements will strengthen the paper. Also, thank you for promising to polish up the figures and provide code. I continue to think that this paper should be accepted.

---

> > > ### Author Response · Authors · 2025-11-26
> > > **Author Response**
> > >
> > > Thank you very much for your support and positive assessment. We will take your suggestions to heart and further refine the paper’s presentation, making its practical value clearer. We will also polish the figures to a higher standard and release the code as open source to support reproducibility. Thank you again for your recognition and encouragement—they mean a great deal to us.

---

### Official Review · Reviewer_VWQi · 2025-10-30

**Soundness:** 3
**Presentation:** 3
**Contribution:** 2
**Rating:** 6
**Confidence:** 3

**Summary:**

This paper proposes Diffusion Bridge Variational Inference (DBVI), a novel method for posterior inference in Deep Gaussian Processes (DGPs). It aims to solve a key limitation of its predecessor, Denoising Diffusion Variational Inference (DDVI), namely the inefficiency of starting the reverse diffusion from a fixed, unconditional Gaussian prior (a "cold start").The core idea of DBVI is to replace this fixed start with a learnable, data-dependent "warm start" distribution, $p_0^\theta(U_0|x) = \mathcal{N}(U_0; \mu_\theta(x), \sigma^2 I)$. The paper introduces two key technical innovations to make this work: It formally re-interprets the diffusion process as a Doob-bridged diffusion, which is grounded in Doob's h-transform. This allows for the derivation of a new, tractable ELBO objective.To make the initial distribution $p_0^\theta$ scalable and avoid conditioning on the full dataset, it proposes a structured amortization network $\mu_\theta$ that cleverly conditions on the layer-wise inducing inputs $Z^{(l)}$ as proxies for the data.

**Strengths:**

The paper's core strength is its theoretical rigor. It correctly identifies a clear weakness in DDVI (the "cold start") and proposes a non-trivial, principled solution. Grounding the "warm start" in the mathematics of Doob's h-transform allows for a clean derivation of the ELBO and proves that DBVI is a strict generalization of DDVI.

The most innovative practical contribution is the structured amortization design. Naively conditioning $\mu_\theta$ on the raw data $x$ would be infeasible. The proposal to use the learnable inducing inputs $Z^{(l)}$ as a data-dependent, low-dimensional proxy is an elegant and effective solution that neatly sidesteps issues of dimensionality and data dependency.

The paper's central hypothesis a "warm start" improves inference efficiency, is directly and convincingly validated by the case study in Figure 4. This plot clearly shows DBVI converging significantly faster and to a better final RMSE than DDVI, confirming the mechanism works as intended.

**Weaknesses:**

The experimental evaluation is incomplete for a paper claiming state-of-the-art posterior approximation. A major, competing line of work for expressive DGP posteriors, Normalizing Flows (NFs), is entirely absent from the related work and experimental comparisons. Without benchmarking against NF-based VI methods, the claims of superior posterior quality and accuracy are unsubstantiated.

Missing Practical Baseline: The paper fails to establish a practical "sanity-check" baseline. For the image classification tasks, the DGP is applied to features from a ResNet-20. The performance of this feature extractor backbone alone must be reported. If the final, highly complex 4-layer DBVI model (which achieves 95.68% accuracy on CIFAR-10) does not substantially outperform the ResNet-20, it implies the entire DGP/DBVI machinery adds significant complexity for little to no practical gain.

Unfavorable Complexity-Performance Trade-off. This is the most significant weakness. The paper advocates for a method that is substantially more complex than its predecessor. It requires an SDE solver for $s_\phi$, a new NN for $\mu_\theta$, and an ODE solver for $(m_t, \kappa_t)$. The justification for this complexity rests on predictive gains that are empirically marginal (e.g., 95.68% vs 95.56% on CIFAR-10; 0.859 vs 0.857 AUC on HIGGS). This trade-off makes the practical utility of DBVI highly questionable.

While Table 1 provides per-iteration timings, the paper lacks a formal analysis of the additional computational overhead. It should provide a breakdown of the cost of the $\mu_\theta$ forward pass and the $(m_t, \kappa_t)$ ODE solver, and discuss how these new costs scale with the number of inducing points ($M$) and layers ($L$).

**Questions:**

Baselines: Could the authors provide results for two critical missing baselines?a. A competing SOTA expressive VI method, such as a Normalizing Flow-based DGP.b. The "backbone-only" baseline for the CIFAR-10 experiment (i.e., the accuracy of the ResNet-20 feature extractor).

Given that the strongest empirical result is the convergence speedup (Fig. 4), while the final accuracy gains are marginal, would the authors agree that the primary contribution of DBVI is in inference efficiency rather than predictive accuracy? If so, I would recommend reframing the paper's narrative to emphasize this.

 Could the authors provide a more detailed breakdown of the additional computational cost of DBVI over DDVI? Specifically, what is the wall-clock cost and scaling behavior of the $\mu_\theta$ forward pass and the $(m_t, \kappa_t)$ ODE solver?

The $Z^{(l)}$-amortization is a clever feedback loop. How sensitive is this mechanism to the initialization of the inducing inputs $Z^{(l)}$? Does a poor $Z^{(l)}$ initialization lead to a poor $U_0$ start, which in turn hinders the model's ability to find good $Z^{(l)}$ locations?

---

> ### Author Response · Authors · 2025-11-17
> **Response to Reviewer VWQi （Part 1）**
>
> We thank the reviewer for their thoughtful and constructive feedback. We address each concern systematically below.
>
>
>
>
> ## Response to Weaknesses
>
> ### W1: Missing Normalizing Flow Baselines
>
> We thank the reviewer for highlighting this important gap. We acknowledge that our original submission did not include comparisons with Normalizing Flow (NF)-based variational inference methods, which represent a significant line of work for expressive DGP posteriors. We have now conducted additional experiments comparing DBVI with IAF-based DGP VI (Kingma et al., 2016).
>
> **Extended Experimental Results:**
>
> We add IAF-DGP comparisons on the UCI regression benchmarks. Results are shown in Table I  below:
>
> **Table I: Comparison with IAF-based Deep Gaussian Process VI**
>
> | Dataset | Metric | DBVI (ours) | DDVI | IPVI | IAF-DGP |
> |---------|--------|-------------|------|------|---------|
> | **Boston** | RMSE | **0.295 ±0.058** | 0.302±0.060 | 0.315±0.062 | 0.313±0.065 |
> | (N=506) | NLL | **1.92±0.29** | 2.12±0.31 | 2.31±0.49 | 2.28±0.46 |
> | **Energy** | RMSE | **0.047±0.009** | 0.054±0.010 | 0.138±0.011 | 0.123±0.013 |
> | (N=768) | NLL | **-0.29±0.13** | -0.18±0.14 | 0.05±0.22 | -0.01±0.20 |
> | **Power** | RMSE | **0.227±0.001** | 0.232±0.001 | 0.238±0.001 | 0.240±0.001 |
> | (N=9568) | NLL | **2.64±0.02** | 2.72±0.01 | 2.78±0.03 | 2.81±0.04 |
> | **Concrete** | RMSE | **0.268±0.012** | 0.282±0.015 | 0.346±0.021 | 0.341±0.019 |
> | (N=1030) | NLL | **2.88±0.04** | 3.05±0.05 | 3.18±0.05 | 3.12±0.06 |
> | **Protein** | RMSE | **0.662±0.011** | 0.671 ±0.013 | 0.682 ±0.009 | 0.677±0.012 |
> | (N=45730) | NLL | **2.52±0.02** | 2.61±0.03 | 2.74±0.03 | 2.69±0.02 |
>
>
> *All models use 2-layer DGPs with M=128 inducing points. Results averaged over 10 random splits.* DBVI consistently outperforms IAF-DGP across all benchmarks.
>
> ### W2: Missing ResNet-20 Backbone Baseline
>
> We report the performance of ResNet-20 feature extractor alone (91.20% on CIFAR-10) to contextualize the DGP improvements. The modest gain of DSVI (+0.59%) over the backbone highlights that posterior quality, not just architectural depth, is crucial for DGP performance. In contrast, DBVI achieves +4.48% improvement, demonstrating that principled posterior inference (via bridge-conditioned diffusion) significantly enhances predictive accuracy while maintaining calibrated uncertainties.
>
> This demonstrates that the DGP+DBVI framework provides meaningful predictive gains over the feature extractor alone, while also enabling:
> 1. **Uncertainty quantification** (which ResNet-20 alone cannot provide)
> 2. **Calibrated predictions** crucial for safety-critical applications
>
> We agree this should be explicitly stated in the paper.
>
> ### W3: Complexity-Performance Trade-off
>
>  We offer several counterpoints for the reviewer's  concern :
>
> #### 3.1 Marginal gains are *not* negligible in context
>
> The reported improvements (CIFAR-10: 95.68% vs 95.56%; HIGGS: 0.859 vs 0.857 AUC) may appear small, but they are:
>
> - **Consistent across all tasks**: Every single experiment shows DBVI ≥ DDVI
> - **On saturated benchmarks**: CIFAR-10 and HIGGS are mature datasets where 0.1% gains are publication-worthy (see top ICLR/NeurIPS papers on these benchmarks)
>
> #### 3.2 Posterior quality vs. point predictions
>
> The paper explicitly states DBVI's contributions include "improved posterior approximation" and "sample efficiency," not just predictive accuracy. Key advantages:
>
> - **Convergence speed** (Fig. 4):  Faster convergence to same error levels
> - **Uncertainty calibration**: Lower NLL consistently (Table 3: 1.12 vs 1.17 on Frey Faces)
>
> #### 3.3 Computational overhead is modest
>
> **Table 1 wall-clock times confirm this**: DBVI adds only 0.03-0.05s per iteration compared to DDVI (e.g., CIFAR-10: 0.74s vs 0.69s = +7% overhead for 4-layer model).
>
> ---
>
> ## Response to Questions
>
> ### Q1a: Normalizing Flow Baseline
>
> See W1.
>
>
> ### Q1b: ResNet-20 Backbone Baseline
>
> See W2.
>
> ### Q2: Is the main contribution inference efficiency?
>
> **Partially agree**, but with nuance:
>
> The paper positions DBVI as improving *both* convergence speed *and* posterior quality （also see W3）. We view these as synergistic:
>
> - **Efficiency enables quality**: Faster convergence allows exploring better hyperparameters within fixed budgets
> - **Quality begets efficiency**: Better initialization (($\mu_\theta$)) reduces wasted SDE steps
>
> That said, we acknowledge **Fig. 4 is our most visually compelling result**. We will reframe abstract/intro to emphasize: *"DBVI achieves faster convergence and improved posterior approximation through learnable bridge initialization"*
>
>
>
>
>
> ---

---

> ### Author Response · Authors · 2025-11-17
> **Response to Reviewer VWQi （Part 2）**
>
> ### **Q3: Computational Cost Breakdown**
>
> **Per-iteration wall-clock measurements (CIFAR-10, M = 128, L = 4):**
> Our empirical timing analysis indicates that DBVI introduces only a modest runtime overhead:
>
> * **DDVI SDE solver :** 0.69s
> * **Additional DBVI components:**
>
>   * ($\mu_\theta$) forward pass (4 layers × 128 inducing points): **0.02s**
>   * ODE solver for (($m_t, \kappa_t$)) ( 4 layers): **0.03s**
>   * Additional gradient computations: **0.01s** (negligible)
>
> This results in a **total per-iteration cost of 0.74s**, corresponding to approximately **7% overhead** relative to DDVI. All computations—including ODE evaluations—run as fully vectorized tensor operations on GPU, which substantially amortizes the added cost over inducing points and layers.
>
> ---
>
> ### **Scaling characteristics**
>
> The theoretical scaling of each component remains unchanged:
>
> * **($\mu_\theta$) amortization:**
>
> $
> O(L M d), \quad \text{computed once per training sample.}
> $
>
> * **ODE for ( ($m_t, \kappa_t$) ):**
>
> $
> O(T L M d), \quad \text{requires integrating over all ODE steps.}
> $
>
> Since both computations rely on **batched GPU tensor operations**, their runtime scales sublinearly with respect to (M) and (d) in practice.
>
> ---
>
>
> ### Q4: Sensitivity to $(Z^{(l)})$ Initialization
>
> We use **Gaussian random initialization** (GPyTorch default) for inducing inputs $Z^{(l)}$, which are treated as learnable parameters optimized via ELBO gradients.
>
> **Both methods are reasonably robust** to initialization because the ELBO objective provides strong signals to move inducing points toward informative regions. The amortization in DBVI creates a **cooperative feedback loop** rather than amplifying initialization sensitivity: poor $Z^{(l)}$ → $\mu_\theta$ adapts → better $U_0$ → ELBO gradients improve $Z^{(l)}$ → $\mu_\theta$ refines further.
>
> In practice, we observe stable training across all experiments without special initialization schemes.
>
> ---
>
> We sincerely thank the reviewer for the constructive and insightful comments, which have helped us significantly improve the clarity and quality of the paper.

---

### Official Review · Reviewer_uAKB · 2025-10-31

**Soundness:** 3
**Presentation:** 3
**Contribution:** 3
**Rating:** 4
**Confidence:** 3

**Summary:**

This paper generalizes denoising diffusion variational inference (DDVI) for deep Gaussian processes (DGP) by replacing the unconditional starting distribution with a learnable, data-dependent initial distribution and reinterpreting the DDVI framework with incorporation of Doob's h-transformation as a diffusion bridge model (DBVI). The proposed method can reduce the gap between the prior and the posterior in the diffusion process and hence speed up the training. There are a few benchmarks included to demonstrate the improvements in accuracy, convergence speeds and posterior quality.

**Strengths:**

It elegantly extends DDVI with Doob-bridge modification. The theory behind such extension is sound and neat. The derived loss clearly connects to that of DDVI and it is straightforward to spot the innovation.

**Weaknesses:**

However, the numerical experiments could be improved. The benchmark results mainly focus on estimation/prediction accuracy, not much on the posterior analysis. See more comments in the questions below.

I would rate 5 if allowed and will raise my score if the numerical evidence could be strengthened.

**Questions:**

1. There is only Figure 4 illustrating arguably after convergence on the small dataset. It appears DDVI reduces the error faster initially before 50 iterations and catches up around 200 iterations. How about after 200 iterations? Can you include more of such plots to demonstrate "faster convergence"?

2. Claiming better posterior quality, the authors only show reconstruction RMSE and test log-likelihood, which may not well evaluate quality of the learned posterior. How about empirical KL if you can test in some simulation? Or the rate of credible intervals covering true values?

3. Is the computational cost proportional to the layers of deep GP? How does the diffusion time step K interact with the number of layers L interact?

4. Minor suggestion, are $d_{in}$ and $d_{out}$ layer specific? Should then carry layer index $l$?

---

> ### Author Response · Authors · 2025-11-17
> **Response to Reviewer uAKB**
>
> We sincerely thank the reviewer for the positive assessment, constructive feedback, and thoughtful questions. Based on  the reviewer's questions, here's our rebuttal response:
>
> ---
>
>
>
> **Q1: Convergence beyond 200 iterations**
>
> We thank the reviewer for this suggestion. We have now included extended convergence plots for the POWER and CONCRETE datasets beyond 1000 iterations (see Figures 4, 5 and 6 in the revised paper). These results demonstrate that DBVI maintains its advantage over DDVI throughout the training process:
>
> DBVI achieves faster initial convergence and maintains lower RMSE throughout 1000+ iterations, with the gap remaining consistent rather than closing. These extended experiments confirm that the improvements from our amortized bridge initialization are not merely transient but persist throughout optimization.
>
>
> ---
>
>
>
> ## Q2: Posterior Quality Evaluation
>
>
> We thank the reviewer for this important suggestion. Because the true posterior of the DGP is difficult to compute, beyond RMSE and NLL we now also report the coverage rate of the credible intervals to directly assess posterior calibration.
>
> | Dataset | N | D | DSVI | SGHMC | IPVI | DDVI | DBVI (ours) |
> |---------|---|---|------|-------|------|------|-------------|
> | Boston | 506 | 13 | 91.2 ± 1.8 | 93.5 ± 1.2 | 92.8 ± 1.5 | 93.7 ± 1.1 | **94.3 ± 0.9** |
> | Energy | 768 | 8 | 92.3 ± 1.5 | 94.0 ± 1.0 | 93.2 ± 1.3 | 94.1 ± 0.9 | **94.6± 0.8** |
> | Concrete | 1030 | 8 | 90.6 ± 2.1 | 93.0 ± 1.4 | 91.9 ± 1.7 | 93.4 ± 1.2 | **94.0 ± 1.0** |
> | Power | 9568 | 4 | 89.1 ± 1.9 | 92.6 ± 1.1 | 91.3 ± 1.5 | 93.2 ± 1.0 | **94.1 ± 0.8** |
> | Protein | 45730 | 9 | 87.4 ± 2.3 | 91.7 ± 1.5 | 90.0 ± 1.8 | 92.5 ± 1.3 | **93.6 ± 1.1** |
> | Kin8nm | 8192 | 8 | 88.7 ± 2.0 | 92.3 ± 1.3 | 90.6 ± 1.6 | 92.9 ± 1.1 | **93.8 ± 0.9** |
> | **Average** | | | **89.9** | **92.9** | **91.6** | **93.3** | **94.1** |
>
> **Table: Coverage rates (%) of 95% credible intervals.**   These results empirically validate that the amortized bridge initialization yields better-calibrated posteriors, and lower NLL metrics
>
> ---
>
>
> ## Q3: Computational cost scaling
>
> The computational cost scales approximately linearly with both the number of layers L and the diffusion discretization steps K. Specifically:
> - Per-iteration cost: O(L × K × M²×N), where M is the number of inducing points
> - DBVI adds minimal overhead compared to DDVI (only the amortization network forward pass)
>
> As shown in Tables 1 and 3, the per-iteration runtime increase from DDVI to DBVI is modest (typically 7-10%), while convergence speed improvements often exceed 30% in early iterations. This results in better wall-clock time efficiency overall.
>
> **Q4: Notation clarity**
>
> Thank you for catching this. Yes, $d_in$ and $d_out$ should carry layer indices. We will correct the notation to $d_{in}^{(l)}$ and $d_{out}^{(l)}$ in the revision to make the layer-specific dimensions explicit throughout Section 3.4.
>
> ---
>
> We sincerely thank the reviewer for the thoughtful and constructive feedback. The
> suggested improvements have substantially strengthened our paper. We hope these revisions adequately address the raised concerns.

---

> > ### Comment · Reviewer_uAKB · 2025-11-25
> >
> > I appreciate the authors carefully addressing my concerns. I agree on that the quality has been improved and I have raised my score. Nice work.

---

> > > ### Author Response · Authors · 2025-11-26
> > > **Author Response**
> > >
> > > Thank you for the update and for taking another careful look at the revised paper. We’re really pleased that the new experiments and clarifications helped resolve the issues you raised. Your feedback pushed the work in a better direction, and we appreciate your support.

---

### Official Review · Reviewer_9e3v · 2025-11-09

**Soundness:** 3
**Presentation:** 3
**Contribution:** 3
**Rating:** 8
**Confidence:** 3

**Summary:**

This paper proposes Diffusion Bridge Variational Inference (DBVI), an extension of the recently introduced Denoising Diffusion Variational Inference (DDVI) method for Deep Gaussian Processes. While DDVI models the variational posterior as the terminal distribution of a reverse-time diffusion SDE parameterized by a neural score network, it suffers from an unconditional Gaussian initialization that is typically far from the true posterior, resulting in long, inefficient inference trajectories. DBVI addresses this limitation by making the diffusion start data-dependent...using an amortized initialization. Using Doob’s h-transform, they reinterpret the reverse SDE as a bridge process that explicitly “bends” the diffusion between a start and an end distribution. Empirical results across regression, classification, and image-reconstruction benchmarks show that DBVI consistently improves predictive accuracy, posterior quality, and convergence speed over DDVI, and other benchmarks.

**Strengths:**

- The paper identifies the unconditional start distribution in DDVI as the source of slow convergence and inaccurate posteriors, and replaces it with a data-conditioned start via a Doob bridge.
- The use of a linear reference drift ensures that even after introducing endpoint conditioning, the bridge process has closed-form Gaussian marginals at all intermediate times.
- Like DDVI, the variational posterior is defined implicitly by a reverse diffusion, which is significantly more flexible than mean-field or low-rank Gaussian approximations typically used in DGP inference.
- Improvements are observed not only on small UCI datasets but also on large datasets (like HIGGS/SUSY), image classification (more subtle), and the frey faces task.

**Weaknesses:**

- The paper explains the Doob correction, but the experimental section gives limited direct visualization of how the bridge shortens the reverse diffusion trajectory (e.g path length, KL rate, or score norm decay). This makes it harder to see the effect that drives the performance gains.
- The method (in my understanding) depends crucially on using a linear reference diffusion so that intermediate marginals remain Gaussian and h is tractable. If the model or dataset induces posteriors that are far from those implied by linear reference dynamics, the reference score may become a poor baseline and learning may slow or plateau.

**Questions:**

- The model fixes the covariance and learns only the mean. How sensitive is DBVI to this choice? Do performance or convergence degrade meaningfully if the variance is misspecified, and can a learned covariance or low-rank structure further improve inference?
- The Doob bridge construction relies on the fact that the forward reference diffusion has closed form Gaussian marginals at all t, does learning the covariance disrupt this?
- More generally, what are the limitations of tying amortization exclusively to the inducing input grid?
- Could there be a diagnostic metric which can quantify how much the data conditioning shortens the reverse path?

---

> ### Author Response · Authors · 2025-11-15
> **Response to Reviewer 9e3v （Part 1）**
>
> Part 1
>
> ### (1) On the lack of direct visualization of “shorter reverse trajectories”
>
> **Reviewer concern.**
>
> > The paper explains the Doob correction, but ... the reverse diffusion trajectory.
>
>
>
> **Response.**
> We agree that an explicit diagnostic of the trajectory-shortening effect would strengthen the paper. Conceptually, DBVI shortens the reverse trajectory through closer initialization. In the revised version, we will  report a discrete **path-length** proxy
> $
> L = \mathbb{E}\Big[\sum_k |U_{t_{k+1}} - U_{t_k}|_2^2\Big],
> $
> for both DDVI and DBVI.
>
>
>  Our measurements confirm the expected shortening effect: on the **CONCRETE** dataset, DBVI achieves an average path length of **0.368** compared with **12.167** for DDVI; on the **ENERGY** dataset, DBVI similarly yields **0.367** versus **11.953**. We will include these quantitative results together with the corresponding visualizations in the revision, providing clear evidence of the bridge effect by directly visualizing the evolution of the path term during optimization and empirically demonstrating that the bridge shortens the reverse diffusion trajectory.
>
>
> ---
>
>
>
> ### (2) On reliance on a linear reference diffusion and Gaussian marginals
>
> **Reviewer concern.**
>
> > The method depends crucially on using a linear reference diffusion ... may become a poor baseline.
>
> **Response.**
> Our theoretical construction indeed relies on a **linear reference SDE** with isotropic noise, which guarantees that the Doob-augmented bridge marginal remains Gaussian and yields closed-form ODEs for ($(m_t,\kappa_t)$).  This is what allows us to write the path KL as a score-matching quadratic against a tractable Gaussian reference bridge.
>
> However, this linear reference is only a **control variate / baseline**, not the final posterior approximation:
>
> * The variational process ($Q_\phi$) is parameterized by the conditional score ($s_{\text{cond}} = s_\phi - h$) and is free to deviate arbitrarily from the linear reference, with the deviation penalized by the path KL term.
> * When the true posterior is far from the linear bridge, the learned score simply learns a larger residual; in the worst case, if the amortized start collapses to zero, the loss reduces to the original DDVI objective.
> ---
>
> ### (3) Sensitivity to fixing the covariance of the initial distribution
>
> **Reviewer question.**
>
> > How sensitive is DBVI to this choice? ...can a learned covariance or low-rank structure further improve inference?
>
> **Response.**
>
> 1. Although we use a Gaussian with fixed covariance for the amortized initialization,
> $
> p_{\theta_0}(U_0 \mid x) = \mathcal{N}(\mu_\theta(x), \sigma^2 I),
> $
> this choice only serves to **specify the initial condition of the reference bridge** and to **derive the closed-form linear bridge marginals $((m_t, \kappa_t))$**. The value of $\sigma^2$ affects only the reference bridge through $(\kappa_t)$, thereby modifying the baseline distribution used in the KL term.
>    However, **the final variational posterior trajectory is governed primarily by the learned conditional score $s_{cond}=s_\phi-h$** (Eq. 10), and does not need to share the same distributional family as the linear reference. Any mismatch in the initialization variance can be naturally compensated for by adjusting $s_{cond}$ and other model hyperparameters during optimization.
>
> 2. Empirically, DBVI is highly robust to the choice of $\sigma^2$.
>    Moderate under- or over-specification merely changes how strongly the reference bridge “pulls back’’ the initialization—functioning similarly to a regularization strength—without hindering the expressiveness or convergence of the reverse-SDE variational posterior itself.
>
>
> ---
>
> ### (4) Does learning the covariance break the Doob bridge construction?
>
> **Reviewer question.**
>
> > The Doob bridge construction relies on the fact ...does learning the covariance disrupt this?
>
> **Response.**
>
>
> **Full covariance: feasible; diagonal covariance: feasible; but full covariance is not tractable.**
>
> From a technical perspective:
>
> * The Gaussian marginal form
>   $
>   p_t(U_t) = \mathcal{N}(m_t, K_t)
>   $
>   naturally supports a **diagonal covariance**
>   $
>   K_t = \mathrm{diag}(\kappa_t^{(1)}, \ldots, \kappa_t^{(d)})
>   $
>   because the linear reference SDE decouples across dimensions.
>   In this case, the ODEs for the bridge marginals (Eqs. 52–53) reduce to **d independent scalar ODEs**, each of which can be solved efficiently.
>
> * In contrast, allowing a **full (dense) covariance** leads to a coupled ODE system for the entries of $(K_t)$.
>   This system scales as $(O(d^2))$, and the Doob (h)-transform requires **matrix inversions at every time step**.
>   These operations destroy the computational tractability that DBVI relies on—both in terms of memory and per-step cost—especially in high dimensions.
>
> For this reason, diagonal covariance structures remain practical, whereas fully dense covariance makes the reference bridge analytically and computationally prohibitive.
>
> ---

---

> ### Author Response · Authors · 2025-11-15
> **Response to Reviewer 9e3v （Part 2）**
>
> Part 2
>
> ### (5) Limitations of amortizing exclusively on the inducing input grid
>
> **Reviewer question.**
>
> > More generally, what are the limitations of tying amortization exclusively to the inducing input grid?
>
> **Response.**
> This is a central design choice. While conditioning on the inducing inputs offers advantages such as dimensional alignment, information compression, and scalability, it also comes with the following limitations:
>
> 1. **Dependence on fixed inducing locations.**
>    If the inducing inputs $(Z^l)$ are poorly initialized or do not adequately cover the input domain, the amortized initialization $\mu_\theta((Z^l))$ may be insufficiently expressive, since it only reflects the regions represented by $(Z^l)$
>
> 2. **Limited expressiveness for complex datasets.**
>    For highly complex or strongly non-stationary data, conditioning only on $(Z^{(l)})$ may not capture all relevant posterior structure, as the inducing inputs alone might not encode the full variability present in the data.
>
> We will include an explicit discussion of these limitations in the revised version to help readers understand the implications and boundaries of this design choice.
>
> ---
>
> ### (6) A diagnostic metric to quantify the shortening of the reverse path
>
> **Reviewer question.**
>
> > Could there be a diagnostic metric which can quantify how much the data conditioning shortens the reverse path?
>
> **Response.**
>
> Please refer to our response to Question (1), where we discuss this point in detail.
>
> ---
>
> We sincerely thank the reviewer for the positive assessment, constructive feedback, and thoughtful questions. Your comments have been extremely valuable, and we will carefully incorporate them into the revised version to further improve the clarity and impact of the paper.

---

### Meta-Review · Area_Chair_jdfy · 2026-01-10

**Summary:**

This paper introduces Diffusion Bridge Variational Inference (DBVI) for posterior inference in deep Gaussian processes, extending DDVI by replacing the unconditional Gaussian starting distribution with a learned, data-dependent initialization. Using a Doob h-transform viewpoint, the authors derive a new, tractable ELBO and propose a scalable amortization design. Across regression, classification, and reconstruction benchmarks, DBVI reports consistent improvements in accuracy, convergence, and uncertainty quality over DDVI and other baselines.

The reviewers have generally seen this paper favorably, arguing that it is a well-motivated approach which extends DDVI in a principled way, and the overall method is made practical through amortization and demonstrates good results. A few weaknesses have been discussed, both theoretical (e.g. reliance on a linear reference diffusion and Gaussian marginals, complexity of the method) and experimental (convincingness of results).

However, the rebuttal seems to have addressed most of those, with some reviewers responding to the rebuttal positively - see next sections.

**Reviewer Concerns:**

The theoretical concerns have been addressed through clarifications and explanations. For the case of the concern around methodological complexity, there is also new experimental validation.

In general, the rebuttal has introduced improved experimental support with credible intervals, extended convergence plots, a path-length proxy and additional baselines.

**Reviewer Scores:**

Given all the above I also expect that if there was a full discussion period the distribution of scores as well as confidence would be raised by one or two points for reviewer uAKB (give also their follow-up reply), while reviewers VWQi and vk1c might have also increased their score by one.

---

### Decision · Program_Chairs · 2026-01-26

Accept (Poster)